

# The role of tidal modulation in coastal flooding on a micro-tidal coast during Central American Cold Surge events

Wilmer Rey[1], Paulo Salles[2,3], E. Tonatiuh Mendoza[2,3], Alec Torres-Freyermuth[2,3], Christian M. Appendini[2,3].

[1]Programa de Maestría y Doctorado en Ingeniería, Universidad Nacional Autónoma de México, DF 04510, Mexico
[2]Laboratorio de Ingeniería y Procesos Costeros, Instituto de Ingeniería, UNAM, México. Puerto de abrigo s/n, 92718 Sisal, México.
[3]Laboratorio Nacional de Resiliencia Costera (LANRESC), CONACyT, Yucatán, Mexico.

*Correspondence to:* Paulo Salles (psallesa@iingen.unam.mx)

**Abstract.** Coastal flooding in the Yucatan Peninsula is mainly associated with storm surge events triggered by high-pressure cold fronts systems passing through the Gulf of Mexico. To assess coastal flood hazards, this study uses a thirty-year water level hindcast, and considers the contribution of wave setup and the role of tidal hydrodynamics. To diagnose the mechanisms controlling the water levels, extreme sea level occurrence probability at Progreso Port was performed to identify the two worst storms in terms of maximum residual tide (Event A), and maximum water level (Event B). Numerical results suggest that during Event A the wave setup contribution reaches 0.35 m at the coast and 0.17 m inside the back-barrier lagoon, while these values are smaller for Event B (0.30 m and 0.14m, respectively). Besides, numerical results of the effect of the astronomical tidal phase on the wave set-up and the residual sea level show that: (i) the wave set-up is tidally modulated and contributes up to 14% to the extreme water levels at the inlet, (ii) the residual tide is larger (smaller) during near-low (high) or receding (rising) tide, and (iii) maximum flooding occurs when the storm peak coincides with rising or high tide, despite micro-tidal conditions.

## 1 Introduction

The Yucatan Peninsula Coast is prone to coastal flooding owing to both its geographical location and geological characteristics. On one hand, extreme meteorological phenomena such Central American Cold Surge (CACS) and tropical cyclones are ubiquitous in this area (Meza-Padilla et al., 2015; Posada-Vanegas et al., 2011; Rey et al., 2016). On the other hand, the region is characterized by a wide and shallow continental shelf (CS), a low-lying coast, the presence of semi-enclosed back-barrier water bodies (lagoons, ports, wetlands). Thus, forcing agents and coast characteristics enhance the vulnerability in this region.

Tropical cyclones occur during summer months and are responsible of the worst coastal flooding events in the Yucatan Peninsula. Nevertheless, they are less frequent in the northern Yucatan coast that has been exposed to only 25 cyclones in the past 150 years (until 2001), i.e., an average of only 0.16 events per year (Rosengaus-Moshinsky et al., 2002), and two cyclones affected the northern Yucatan coast from 2002 to present (0.15 events/year). On the other hand, most frequent CACS (locally knows as Nortes) occurring during late fall and winter present an annual mean of 16 events (Reding, 1992). Following Reding (1992) and Schultz et al. (1998), the present study defines a CACS as an anticyclone movement of a cold mass of air originated in North America, which penetrates equatorward to at least 20º N latitude.

Several efforts have been made to determine the frequency of the CACS in the Gulf of Mexico -GoM- (DiMego et al., 1976; Henry, 1979; Reding, 1992; Schultz et al., 1998).The frequency and penetration of the cold fronts over the GoM towards the equator is directly related with the topographic characteristics, position, strength, and amplitude of the circulation on mid-latitudes (DiMego et al., 1976). Characteristics prior to frontal pass over GoM are veering wind, rising temperature, falling pressure and increase of humidity (Reding, 1992). This author also determined the presence and frequency of CACS that have passed through the Yucatan Peninsula from 1979 to 1990, using the hourly surface wind database and images obtained from the GOES satellite imagery





(infrared and visible) based on two criteria: decrease in daily maximum temperature greater or equal to 4 ºC during 48 consecutive hours in Merida (capital of the Yucatan State), and sustained northerly winds (more than 24 hours and coming from the azimuthal 300º - 30º sector). This study found that the average duration of CACS is between 3 and 6 days, but may last up to 13 days. Schultz et al. (1998) followed the methodology applied by Reding (1992) to determine the frequency of cold fronts employing radiosonde

data for the months of October through March from 1957 to 1989. This work identified 363 events and a mean yearly frequency of 11 events. More recently, López-Méndez (2009) defined a CACS as an event characterized by having a pressure (reduced at sea level) higher than 1,020 hPa in the geographical coordinates 30°N and 100°W, that simultaneously has associated wind speeds higher than 12m/s at the 20°N and 93.75°W coordinates, and occur between September and April although no number of events are reported in this work.

The northern coast of the Yucatan Peninsula is the first landmass to interact with the CACS after they have crossed the GoM, where the magnitude of the surface wind speed can reach up to 30 m.s$^{-1}$, according to measurements from the National Data Buoy Center (NDBC) stations and Schultz et al. (1997). The shear stress on the sea surface, the fetch and the duration of the events generate strong waves in the southern GoM, as well as storm surges in the Mexican GoM coast, including the northern Yucatan Peninsula coast. The storm surge is induced by the wind shear stress on the sea surface and perturbations on the atmospheric

pressure (Lin and Chavas, 2012). Since the effect of the pressure is relatively small, particularly in high-pressure atmospheric systems as CACS, the storm surge is mainly due to the shear stress of the wind and the presence of a land mass, especially in shallow waters in the coastal zone (Flather, 2001). Depending on the shape of the basin (concave or convex), the direction and duration of the incident wind, the coastal bathymetry as well as the CS extent, a higher or smaller storm surge will occur. Additionally, seasonal variations of the mean sea level and pressure gradients induced by currents are other factors (e.g., steric

basin-scale anomaly, astronomical annual tidal component) that may cause sea level increase, with maxima for this region in September-October (Zavala-Hidalgo et al., 2003).

According to Merz et al. (2007), flood hazard, is defined as the probability of the induced potential damage caused by a flood in a determined area and period, depends on several factors such as the maximum water level reached, the flux velocity, the flood duration, the sea level rising speed, and the flood frequency. However, wave set-up can be important for the accurate estimation

of the extreme water levels during CACS events. During the wave breaking process the kinetic energy is converted, to a great extent, to a quasi-steady potential energy, generating a water surface gradient to balance the onshore component of the momentum flux due to the presence of waves (Dorrestein; Longuet-Higgins and Stewart, 1963). Consequently, increase of the water level along the shoreline (Dodet et al., 2013; Smith et al., 2000) as well as wave-induced currents are generated. Besides, when inlets or port entrances are present, as in the case in the northern Yucatan coast and in particular in Progreso, these processes play an

important role in modifying the inlet and lagoon hydrodynamics and morphodynamics regularly forced only by the astronomical tide and fresh water input from springs due to cracks in the large confined aquifer. Moreover, given that wave breaking in the inlet ebb shoal or further inside depends on the water depth, both the wave breaking-induced acceleration and the wave set-up are tidally modulated (Olabarrieta et al., 2011). However, due to the computational time cost for numerical modeling the wave set-up is commonly neglected, especially in real-time forecasting of hurricane storm surge when prompt simulation results are required for

preparing evacuation plans. Lin et al. (2012) simulated a set of over 210 extreme surge events with the ADCIR model coupled with the SWAN wave model and found that the wave set-up accounted for less than 1.5% of the surge for four locations around New York Harbor. They suggested that the smallness of the wave set-up in that area may be related to the fact that large ocean waves break before entering the New York Harbor, and also because the near-shore wave breaking may not be captured by the larger scale computational mesh of models. They also found that run time for the surge-wave coupled simulations was one order

of magnitude larger than for the simulations accounting only for the surge. For the Yucatan coast, flood hazard studies have not





taken into account the wave set-up contribution and hence the present work aims to investigate its relative importance in coastal flooding during CACS events.

During CACS passage, both erosion and flooding processes occur, but given the low lying coast, flooding represents a higher hazard (Mendoza et al., 2013), and given their relatively frequent occurrence, their socio-economic impact is high. For instance,

during CACS passage both port and oil industry activities are affected in the southern GoM, which translates in economic loss, as was the case of Cold Front # 4 on October 23, 2007, which caused major flooding and coastal structure damage in Villahermosa, in the nearby state of Tabasco, and whose total economic loss was estimated in approximately 2.45 billon usd (López-Méndez, 2009). Another case of flooding induced by a CACS surge occurred on November 28, 2013 in the northern coast of Yucatan, where Dzilam de Bravo community was flooded, causing damage to around 80 houses and coastal structures(Cob-Chay et al., 2013).

Flood hazard assessments are normally performed based on historic or synthetic flood data (Lin et al., 2010), but since wind reanalysis datasets have become available such as the North American Regional Reanalysis- NARR (Mesinger et al., 2006) and the Climate Forecast System Reanalysis-CFSR (Saha et al., 2010), they have been used to force hydrodynamic models to generate sea level reanalysis. However, given that the wind fields during tropical cyclones are underestimated in these reanalysis (Swail and Cox, 2000), their applicability is not recommended when such severe events occur, but can successfully be used to estimate

the storm surge during CACS. Furthermore, since long records of raw sea level data are scarce for the Northern coast of the Yucatan Peninsula (barely 5 years of good quality historical data is available for the Progreso area: 1979-1984), the sea level hindcast was selected as an alternative, and used to perform a robust CACS-generated extreme sea level analysis and the flood hazard assessment, both presented below.

Given that the CFSR database underestimates wind fields during tropical cyclones, the hydrodynamic model also underestimated

the sea level during these events. The hazard due to hurricane storm surge in the Yucatan coast has been investigated previously (Meza-Padilla et al., 2015; Posada-Vanegas et al., 2011) as well as the hurricane flood risk (Rey et al., 2016) and hence is beyond the scope of the present study. Therefore, the aim of this study is to assess flood hazards caused by CACS in the northern Yucatan Peninsula with emphasis on evaluating the relative contribution of storm surge and wave set-up in Progreso, Yucatan. Furthermore, the numerical model is also employed to investigate the role of the astronomic tidal phase on those processes.

**2 Study area**

The study area comprises the northern coast of the Yucatan Peninsula (Figure 1), which extends from Cancun to Celestun, and more specifically is focused on (i) the town of Progreso, which is the most urbanized, populous and economical important coastal city in the Northern coast of Yucatan, and (ii) the back-barrier Chelem lagoon behind Progreso. The maximum ground elevation in Progreso is 2.1 m above Mexican Geoid GGM06, within a low-lying coast (average elevation in Merida, the capital: 8 m,

approximated 27 km inland). The Peninsula averages ten meters above sea level with only a small prominent Sierra in the center, where maximum altitude reaches 150m (Stringfield and LeGrand, 1974). The northern Yucatan coast is mostly sandy (85% of its length) and 57% is formed by coastal lagoons and barrier islands (Cinvestav, 2007). In terms of hydrogeology, the northernmost section of the fresh-water aquifer of Yucatan is confined near the coast by a thin, flat, nearly impermeable layer probably in formed by a process of sedimentation and precipitation of porefilling cement near the surface of older limestone at the landward margin

of the swamp that extends along the north Yucatan coast (Perry et al., 1989). The presence of $^{14}C$ in the cemented layer support the hypothesis that cementation is an ongoing process in this locality (Bauer-Gottwein et al., 2011; Perry et al., 1989). This coastal nearly impermeable layer acts as a partial dam, impeding the seaward movement of fresh water to extend more than 3 km into the ocean (Perry et al., 1989), although some fractures in this layer produce fresh water springs in the coastal ocean.



The climate in the zone is characterized by three seasons: warm and dry season (March-May), rainy season (June-October) and winter season with occasional showers (November-February) associated with the CACS passage (Schmitter-Soto et al., 2002) . Predominant winds are associated to sea-/land- breezes from the NE/SE that are more frequent and intense during spring (Torres-Freyermuth et al., 2014). On the other hand, during winter months strong northerly winds interact with the maritime tropical wind

from the Caribbean, causing the CACS (Schultz et al., 1998), which are distinct from the northeasterly sea breeze because of its duration (Reding, 1992), atmospheric pressure and air temperature. The mean annual precipitation varies between 444 mm and 1,227 mm, Progreso being located in the driest area of the Peninsula (INEGI, 2002).

In terms of bathymetry and energy dissipation, the karstic CS is exceptionally wide (up to 245 km), shallow and on average with a mild slope -1/1000- (Enriquez et al., 2010) in which, on one hand, the wind wave energy is low (Lankford, 1976) near the coast

due mainly to bottom friction dissipation, and on the other hand the storm surges are amplified. Long-term wave climate analysis in this zone was performed by Appendini et al. (2013) by means of a thirty-year wave hindcast for the GoM and Caribbean Sea, showing that the CS dissipates storm waves from distances far offshore of the order of tens of kilometers. This, together with the mixed tide with a diurnal dominance and a small neap range of 0.1 m and 0.8 m during spring (Cuevas-Jiménez and Euán-Ávila, 2009), confers to the coastal zone a low energy regime during practically all circumstances (Salles et al., 2013), except at the

eastern part of the Peninsula where the wave energy increases due to a narrower CS (Appendini et al., 2012).

On the other hand, Appendini et al. (2012)reported a net westward transport longshore sediment along the entire northern Yucatan coast, ranging between 20,000 and 80,000 $m^3.y^{-1}$, except west of Holbox, where longshore transport direction is inverted. The dominant westward longshore transport suggests an extremely sensitive shoreline to artificial littoral barriers (e.g., groins and jetties).

## 3 Numerical Model

### 3.1 Mathematical formulation

The HD model used for this study was the MIKE 21, developed by DHI Water & Environment, which resolves the two dimensional shallow waters equations –the depth-integrated incompressible Reynolds Navier-Stokes equations invoking the assumptions of Boussinesq and Hydrostatic pressure (DHI, 2014a) This model has been  successfully used in recent scientific studies (Appendini

et al., 2014; Meza-Padilla et al., 2015; Strauss et al., 2007) Integration of the horizontal momentum equations and the continuity (1) equation over $h = \eta + d$, the following two dimensional shallow water equations are obtained.

$$\frac{\partial h}{\partial t} + \frac{\partial h\bar{u}}{\partial x} + \frac{\partial h\bar{v}}{\partial y} = hS \tag{1}$$

and the two horizontal momentum equations for the x-component (2) and y-component (3), in Cartesian co-ordinates respectively:

$$\frac{\partial h\bar{u}}{\partial t} + \frac{\partial h\bar{u}^2}{\partial x} + \frac{\partial h\bar{v}\bar{u}}{\partial y} = f\bar{v}h - gh\frac{\partial \eta}{\partial x} - \frac{h}{\rho_0}\frac{\partial p_a}{\partial x} - \frac{gh^2}{2\rho_0}\frac{\partial \rho}{\partial x} + \frac{\tau_{sx}}{\rho_0} - \frac{\tau_{bx}}{\rho_0} - \frac{1}{\rho_0}\left(\frac{\partial s_{xx}}{\partial x} + \frac{\partial s_{xy}}{\partial y}\right) + \frac{\partial}{\partial x}\left(hT_{xx}\right) + \frac{\partial}{\partial y}\left(hT_{xy}\right) + hu_sS \tag{2}$$

$$\frac{\partial h\bar{v}}{\partial t} + \frac{\partial h\bar{v}\bar{u}}{\partial x} + \frac{\partial h\bar{v}^2}{\partial y} = f\bar{u}h - gh\frac{\partial \eta}{\partial y} - \frac{h}{\rho_0}\frac{\partial p_a}{\partial y} - \frac{gh^2}{2\rho_0}\frac{\partial \rho}{\partial y} + \frac{\tau_{sy}}{\rho_0} - \frac{\tau_{by}}{\rho_0} - \frac{1}{\rho_0}\left(\frac{\partial s_{yx}}{\partial x} + \frac{\partial s_{yy}}{\partial y}\right) + \frac{\partial}{\partial x}\left(hT_{xy}\right) + \frac{\partial}{\partial y}\left(hT_{yy}\right) + hv_sS \tag{3}$$

where $\eta$ is the surface elevation, $h$ the water depth, $d$ the still water depth, $\rho_0$ the reference density of water, $t$ the time, $x, y$ are the Cartesian coordinates, $g$ the gravitational acceleration, $S$ the magnitude of discharge due to point sources, $u_s, v_s$ are the velocities by which water is discharged into ambient water, $\rho$ the density of water, $\bar{u}, \bar{v}$ the depth averaged velocity in the $x, y$ directions, $p_a$ the atmospheric pressure, $\tau_{bx}, \tau_{by}$ are the components of bottom stress, $s_{xx}, s_{xy}, s_{yx}, s_{yy}$ are the components of





radiation stress tensor, $\tau_{sx}$, $\tau_{sy}$ are the components of surface wind stress, $T_{xx}, T_{xy}, T_{xy}, T_{yy}$ are the components of lateral stress, $f$ the Coriolis parameter. Any variable with an overbar indicates a depth-average value.

The wave model used to compute the wave conditions and associated radiation stresses was the MIKE 21 third generation spectral wave (SW) model, which is adequate for studying spectral wind-wave modeling (Appendini et al., 2013; Appendini et al., 2015;

Strauss et al., 2007). This wave model is based on unstructured meshes, and simulates the growth, decay and transformation of wind-generated waves and swell in offshore and coastal areas. The SW wave model includes the wave growth by action of wind, non-linear wave-wave interaction, dissipation due to white-capping, dissipation due to bottom friction, and dissipation due to depth-induced wave breaking, as well as refraction and shoaling due to depth variations, wave-current interaction and effect of time-varying water depth and flooding and drying (DHI 2014a,b).

The SW wave module is based on the wave action equation where the wave field is represented by the wave action density spectrum $N(\sigma, \theta)$, formulated in terms of the relative angular frequency $\sigma$, and the direction of the wave propagation $\theta$, where the energy density spectrum is related to the wave action density spectrum by

$$N(\sigma, \theta) = E \frac{\sigma, \theta}{\sigma}, \tag{4}$$

This wave model includes two different formulations: directionally decoupled parametric formulation and fully spectral

formulation: the directionally decoupled parametric formulation and the fully spectral formulation. The first is based on a parameterization of the wave action conservation equation, made the frequency domain by introducing the zeroth and the first moment of the wave action spectrum as dependent variables as described in Holthuijsen et al. (1989)and the second formulation is based on the wave action conservation equation as described Komen et al. (1994) and Young (1999), where the directional-frequency wave action spectrum is the dependent variable. Since the fully spectral formulation is used for wave growth, decay and

transformation of wind-generated waves and swell in offshore and coastal areas, this formulation was chosen for this study. The decoupled parametric formulations is more used for small scales transformation applications (less than 10-10 km) and when the developed seas are dominating and swell and combined sea/swell is not important. The wave action conservation equation is written in Cartesian coordinates as

$$\frac{\partial N}{\partial t} + \nabla \cdot (\bar{v} N) = \frac{S_{in} + S_{nl} + S_{ds} + S_{bot} + S_{surf}}{\sigma} \tag{5}$$

where the action density is defined by $N(\bar{x}, \sigma, \theta, t)$, $t$ is the time, $\bar{v} = (C_x, C_y, C_\sigma, C_{\theta,})$ is the propagation velocity of a wave group in the four-dimensional phase space $\bar{x}$, $\sigma$ and $\theta$. $\nabla$ is the four-dimensional differential operator in the $\bar{x}, \sigma, \theta$ -space. The energy source term, $S$, represents the superposition of source functions describing several physical phenomena $S_{in} + S_{nl} + S_{ds} + S_{bot} + S_{surf}$ where $S_{in}$ represents the generation of energy by wind, $S_{nl}$ is the wave energy transfer due to non-linear wave-wave interaction, $S_{ds}$ is the dissipation of wave energy due to white capping, $S_{bot}$ is the dissipation due to bottom friction and $S_{surf}$ is

the dissipation of wave energy due to depth-induced breaking (DHI, 2014b) For more detailed information about source terms, governing equation, time integration and model parameters, readers are referred to (Sørensen et al., 2004) and to the scientific manual documentation for the SW model (DHI, 2014 b)

### 3.2 Model setup

### 3.2.1 Hydrodynamic model setup

The Hydrodynamic model (HD) was used in order to obtain a thirty year currents and sea level hindcast not accounting for waves due to the high computational cost. The domain is shown in Figure 2, where the Yucatan channel boundary was forced with a mean profile of the Yucatan current using the results reported in Abascal et al. (2003). Part of this current variability has been attributed





to mesoscale eddies, which are observed in the eastern Caribbean basin, the Cayman Sea, and western Caribbean passages (Athié et al. 2011). The GoM boundary was forced with astronomic tide varying along the boundary, and was extracted from the tide global model (Andersen, 1995), which represent the major diurnal ($K_1$, $O_1$, $P_1$ y $Q_1$) and semidiurnal tidal constituents ($M_2$, $S_2$, $N_2$ y $K_2$) with a spatial resolution of 0.25 x 0.25 degrees (DHI, 2014d). The model employs the latest 17 years multi-mission

measurments from TOPEX/POSEIDON (phase A and phase B), Jason-1 (Phase A and phase B) and Jason -2 satellite altimetry for sea level residual analysis and the harmonic coefficients have been calculated (DHI, 2014b). Based on these measurments, this tide model has been widely applied in depths greater than 20 m (DHI, 2014d), which is the case of this boundary. The Campeche boundary was considered as open (sea level equal to zero) and the southern boundary (land) was forced with a constant Yucatan aquifer discharge of $2.710^{-4}$ $m^3.s^{-1}.m^{-1}$ reported by Weidie (1985), which did not affect significantly the sea level. On the surface

the model was forced with wind and pressure fields from the *CFSR* database. Appendini et al. (2013) showed that wind reanalysis in the resolution of NCEP/NCAR to NARR is sufficient for wave modeling of CACS over the GoM. It is then assumed that CFSR resolution is adequate. Given that the model used only accepts wind and pressure data varying in space from a regular grid, CFSR wind and pressure fields were linearly interpolated from a T382 Gaussian grid resolution to a regular grid with spatial resolution of 0.3125 degrees. The boundary conditions were treated following the methodology used by Enriquez et al. (2010).

The bathymetry was extracted from the ETOPO1 database and complemented with higher resolution bathymetric data from 9 km long transects every 4 km along the coast. In addition, high-resolution topography (1 m spatial resolution) from a 2011 LIDAR survey of the entire port of Progreso was used (Figure 2). After a calibration process comparing model results with sea level measurements during three CACS events in Progreso, the bottom friction was defined using a constant Manning coefficient of 0.02, which corresponds, according to (Arcement and Schneider, 1989), to the average mean grain size ($d_{50}$) for the Yucatan sand

beaches reported by Mendoza et al. (2013). For the horizontal eddy viscosity (Smagorinsky formulation) a constant coefficient of 0.28 was applied. The wind friction (Cd) was estimated based on the Garratt (1977) formulation modified by Lin and Chavas (2012), and further calibrated in this study.

### 3.2.2 Wave and Coupled model setup

Once the sea level hindcast was performed, two of the most extreme flooding events were identified from this data set. For the two

identified events, the wave setup contribution was taken into account coupling the HD and the spectral wave model (SW), resulting in the coupled HW.

The HW model used the same computational domain (Mesh 1) as the HD model. However, given the unknown swell wave conditions for the ocean boundaries, a wave model was implemented for the entire GoM (Figure 3) to reproduce distant wave climate accurately to be used as forcing at the ocean boundaries of Mesh1 in the coupled HW model. The wave model set-up was

based on the study by Ruiz-Salcines (2013) who calibrated the model for mean and extreme conditions in the Gulf of México and Caribbean Sea.

### 3.3 Datasets and simulated cases

In this study the term "storm surge" is used when referring only to the meteorological contribution to the total sea level; otherwise, we refer to "residual tide", which may contain storm surge, tide-storm surge interaction, harmonic prediction errors, timing errors

(Horsburgh and Wilson, 2007) and coastal aquifer discharge. In other words, the residual tide is the total water level from the sea level hindcast (without wave set-up) minus the astronomical tide.

For the extreme analysis used for flooding purposes, two data types can be applied. On one hand, the assessment can consider only the residual tide, which is the tide that is not readily predictable and thus is what is associated to risk, and on the other hand it may





be important to take into account when – within the astronomical tidal cycle – this extreme residual occurred, since the CACS can hit the Peninsula in any tidal phase. In other words, the phase of the tide may play an important role: if the storm occurs during low tide the flood risk is lower than if it occurs during high tide. Besides, both the storm surge and the wave set-up are tidally modulated (Olabarrieta et al., 2011).

Given the above, two datasets were used in the extreme analysis:

a) D1: The first consisted in identifying the yearly largest residual sea level from the entire thirty-year sea level hindcast at Progreso. The astronomical tide was evaluated by means of a harmonic analysis using the T_Tide program(Pawlowicz et al., 2002), and removed from the output sea level time series.

b) D2: The second consisted in adding the corresponding astronomical tidal contribution to the yearly largest residual sea

levels from (a), to account for the amplitude and phase of the tide as mentioned above.

Firstly, the identification method from Reding (1992)was applied, in order to confirm if the 30 selected events were induced by CACS, concluding that all events were induced by CACS. Next, both datasets – (a) and (b) – were fitted to the Generalized Extreme Values (GEV) distribution probability function (Ho et al., 1976; Jenkinson, 1969).

Furthermore, in order to assess the wave set-up contribution in the flood assessment at Progreso port and inside the back-barrier Chelem lagoon, and given the computational time cost to compute the wave set-up for these 30 events, two events were selected:

- Event A, with the highest residual tide from the hindcast (which happened during receding tide), and
- Event B, with the highest sea surface elevation (which occurred during rising tide).

Event A was selected due to the fact that it was the strongest CACS event in terms of wind intensity and duration of the total thirty-

year hindcast. In fact, Schultz et al. (1997) studied in detail this CACS event due to its exceptional intensity over the Gulf of Mexico, the important role that convection played in the incipient cyclogenesis, the planetary-scale antecedent conditions, and the merger of two short-wave troughs in the westerlies contributing to the extreme cyclogenesis. Bosart et al. (1996) as well as Schultz et al. (1997) called this event "the 1993 super storm cold surge, also known as the storm of the century", which originated over Alaska and western Canada, and brought northerlies exceeding 20 ms$^{-1}$ and temperature decreases up to 15 °C over 24 h into

Mexico and Central America.

During the peak of the storm in Event A, the astronomic tide and residual tide were, -0.35 m and 1.14 m, respectively (i.e., the largest residual tide of the thirty-year sea level hindcast), resulting in a total sea level of 0.79 m. On the other hand, during the peak of the storm of Event B, the astronomic tide and residual tide were +0.44 m and 0.72 m, respectively, resulting in a total sea level elevation of 1.16m, i.e., 0.37m higher than the total sea level during Event A in the coast offshore. Figure 4 presents wind speed

and direction from the 42001 NDBC buoy (see location in Figure 3) during events A and B. Throughout Event A, the predominant wind direction (azimuth) was 315º while for B was 340º, i.e. closer to normal to the coast. Moreover, the maximum wind speed was similar for both events, but the duration of wind speeds higher than 20 m/s was longer for Event A (11 hours compared to 3 hours for Event B). This suggests that the duration of the storm is a predominant factor in the generation of storm surges.

In order to investigate the relative contribution of the (i) storm surge and (ii) wave set-up during the flooding episodes, three

different configurations were implemented for events A and B:

- Configuration 1 (storm surge): the hydrodynamic model was forced on the surface with pressure and wind fields from the CFSR database. Then, only the storm surge contribution was evaluated from this configuration.
- Configuration 2 (wave set-up): the hydrodynamic model was forced only with the radiation stresses obtained from the HW model (wave-current interactions; see section 0), obtaining only the wave set-up contribution and wave-induced currents.

In both configurations the ocean boundaries surrounding Mesh1 were open, and the coastline boundary was closed.



- Configuration 3 (Total Sea Surface Elevation -TSSE-, and total currents): the HW model was used to investigate the contributions from the storm surge, wave set-up, astronomic tide and the Yucatan current, for events A and B only, to assess flood prone areas in Progreso.

- Configuration 4: Due to high computational cost of using the HW model for the complete thirty-year reanalysis, this
configuration 4 considered only the hydrodynamic model. It was used for the reanalysis of the sea level induced by wind, tides and mesoscale currents (see details in section 3.2).

In order to further investigate the hydrodynamics occurring in the Chelem and Progreso vicinity, additional numerical experiments were performed (wave induced currents, wave set-up, storm surge, astronomic tide and currents), as a function of the tidal phase during a CACS passage. Therefore, three numerical experiments were produced, based on Configuration 3 for Event A and varying
the tidal phase:

- Tide Scenario 1 (TS1) high tide.
- Tide Scenario 2 (TS2) receding tide near mean sea level.
- Tide Scenario 3 (TS3) receding tide near low tide (Actual Event A).
- Tide Scenario 4 (TS4) rising tide near mean sea level.

In addition, these four "tide scenarios" were also used with Configuration 4 to study the variation of residual tide as a function of the astronomic tidal phase.

The dashed lines in the upper panel in Figure 5, show the forcing tide for the scenarios mentioned above, which varied in phase but remained with the same amplitude. The first peak of TSSE for TS3 (time t1 in Figure 5) was taken as reference for varying the phase in the other scenarios: TS1 is 12 h ahead of TS3; while the phase shift for TS2 and TS4 were such that the water level is zero
at time t1, during receding and rising tides, respectively. Both t1 (flood) and t2 (ebb) were the times used for assessing the wave set up contribution inside the Chelem lagoon.

Given that the NDBC station 42001 is located 347 miles North from the Yucatan coast, and in order to explore the wind direction behavior between that station and the coast, wind speed and direction were extracted from the CFSR database at 3 different locations: Node C (278 km off the coast), Node B (161 km) and A (80 km) (see Figure 3). The dashed lines in the lower panel in
Figure 5 shows how the wind speed decreases in part because of the roughness change when the wind reaches the landmass. The solid lines denote that the wind direction remained the same for all the nodes during the CACS event.

Besides, Figure 5 shows that the modeled TSSE for TS3 (top panel, right axis, green solid line), which presents two peaks (at times t1 and t2) correlates better with the offshore wind speed time series (bottom panel, left axis) than with the wind at Progreso, suggesting the importance of using a large computational domain for improved accuracy (Blain et al., 1994; Kerr et al., 2013;
Morey et al., 2006). The mesh used in this study (Figure 2) results from a sensitive analysis of the domain size (not shown) to determine the size in which the model adequately reproduced the sea level recorded by a tide gauge at Progreso.

### 3.4 Model validation

The hydrodynamic model results were validated with data from a tidal gauge in Progreso (located at the coordinates-89.6667º W, 21.3033º N), to perform a thirty-year (1979-2008) sea level hindcast for the northern Yucatan Peninsula. For the hydrodynamic
model validation, Figure 6 shows the measured sea level and hydrodynamic simulation results for the two storm events with the greatest residual tide in the 5-years tide gauge record. In general, a good agreement can be seen for the sea surface elevation during the storms the Pearson correlation ranges from 0.78 to 0.87 and the root mean square error (RMSE) ranges from 0.11 to 0.17.

Results from the wave model as well as wind speed and wind direction for Event A is shown in *Figure 7*. Both the CFSR winds and the wave model results (significant wave height and peak period), were validated with measurements from the NDBC stations





42001, 42002, 42003 and 42055 (see locations in Figure 3 ) for events A and B. For instance, during Event A, significant wave heights, Hs, and peak periods, Tp, from the wave model exhibited a good Pearson correlation (0.90 and 0.79, respectively) with respect to the *NDBC* station 42001 data (top panels in Figure 7). Furthermore, a good Pearson correlation (0.91) between the CFSR wind reanalysis and wind measurements from the same *NDBC* station was found (bottom panels Figure 7).

## 4 Results

### 4.1 CACS characterization

The histograms of the main characteristics of the 30 events associated to the annual maximum residual tide are shown in *Figure 8*. It is important to notice that the wind speed (top left panel) and wind direction (bottom right panel) values, as well as the MSL pressure (top right panel) correspond to the time when the maximum residual tide (bottom left panel) occurred, which may not correspond to the maximum values registered during the event for each variable, as shown in *Figure 9* for Event A. The maximum residual tide was 1.14m, which correspond to Event A, but for haft of the 30 events it ranges between 0.4m to 0.5m. Mean sea level (MSL) pressure, wind speed and wind direction where taken from the *CFSR* reanalysis database at the location of the tide gauge in Progreso. Wind directions for all the events are from the north and northwest as expected for the CACS events. MSL pressure oscillates between 1002 and 1021 hPa.

As an example of the CACS effect on the coast, Figure 9 shows the wind speed (top panel, dashed line), pressure (central panel) and daily maximum temperature (bottom panel, dashed line) for Event A from the CFSR database, as well as residual tide from the hydrodynamic model (without waves; top panel, continuous line). It can be seen how from 00:00 on March 13 there is a significant decrease in temperature – 16.7° in 24 hours – (lower panel), an increase in wind speed – 10 m/s in 8 hours – (upper panel), a veering of the wind direction becoming northerly between azimuthal 300 and 30° (lower panel, right axis), as well as an increase in the MSL pressure (9hPa). Therefore, this event contains all the elements to be classified as CACS according to Reding (1992). It can also be seen that the peak of the residual tide occurs after the peak of the wind intensity (roughly 2 h) and after the minimum pressure (roughly 6 h). The maximum daily temperature as well as the wind direction was obtained in Merida in order to follow the Reding (1992) classification method.

### 4.2 Sea level hazard assessment

The first extreme analysis method (see D1 in section 3.3), which consisted in fitting the yearly maximum residual tide at Progreso port to the GEV, is shown in Figure 9, where events A and B are identified. The expression used for the GEV is shown in equation 6).

$$H(x, \mu, \psi, \xi) = exp\left\{-\left(1 + \xi \frac{x-\mu}{\psi}\right)\right\}^{-1/\xi} \qquad \qquad 6)$$

where $\mu$ is the location parameter, $\xi$ is the shape parameter and $\psi$ is the scale parameter. By means of the maximum likelihood estimation method the following function parameters were found $\mu = 0.469$, $\xi = 0.189$ $\psi = 0.105$

The second extreme analysis method (see D2 in section 3.3), which consisted in fitting the sea level associated to the yearly maximum residual tide at Progreso to the GEV, is shown in *Figure 11*. In this analysis, the maximum likelihood estimation method was also used, and the function parameters found were found $\mu = 0.526$, $\xi = -0.295$, $\psi = 0.261$.

It is important to mention that in both cases the extreme analysis was performed using model data from a point situated 2 km offshore (5 m water depth), where the Progreso tide gauge is located. It can be seen from Figure 10 and Figure 11 that:





- When using only the residual tide in the extreme analysis (D1 dataset; see section 3.3), the resulting return periods for events A and B were 67 and 7 years, respectively, given that the residual tide for Event A was larger than for Event B (1.14 m vs 0.72).

- When the astronomic tide is taken into account (D2 dataset), the return periods for events A and B become 3 and 78 years, respectively, given that the sea level including residual and astronomic tides was greater during Event B (1.16 m vs 0.79 for Event A).

This is due to the fact that Event A, whose residual sea level was the largest, happened near low tide, while Event B occurred during spring high tide. Therefore, including the astronomic tide in the extreme analysis is crucial in the estimation of the return period. Moreover, in sections 4.3 and 4.4 below we discuss the importance of not only considering the tidal phase but also the local wind direction when determining the flood risk inside the lagoon, given that the wind setup in this semi-enclosed body can be of major significance, and is crucial since the town of Progreso has its southern limits next to lagoon margins.

### 4.3 Contribution of storm surge and wave set-up to flooding for events A and B.

As mentioned before, the storm surge is induced by the wind shear stress and perturbations on the atmospheric pressure, while the wave set-up depends primarily on the wind waves, and both are affected by the tidal level. Besides, inside the lagoon, the wind set-up (and hence the wind direction and storm duration) and the hydrodynamics at the inlet play an important role. In this sense, Figure 12 presents maps of the highest wind stress and wave height values for events A and B. During Event A (storm shown in Figure 7) wave heights of 4-5 m occurred 2.5 km offshore, and in contrast the same wave height values occurred 5 km offshore for Event B. Moreover, during Event A, the wind direction and wind stress (top, left panel), propagated the waves (top right panel) from the west to an easterly direction inside the lagoon, leading to a large wave height (around 0.9m) within the lagoon in the southeast part of Chelem. On the other hand, during Event B, the wave height (bottom, right panel) inside the lagoon was weaker than Event A, in part due to the weaker wind stress (lower left panel) over the lagoon. These variations caused significant differences for the induced wave set-up and wind set-up inside the lagoon for each event as it will be shown in Figure 13.

The maximum values of storm surge and wave set-up for Events A and B are shown in Figure 13. The top panels (left and right) show model results of the maximum storm surge. Through normal weather conditions there is a predominant littoral current along the coast of the north part of the Yucatan Peninsula from east to west (Enriquez et al. 2010), which is attributed to sea breezes (Torres-Freyermuth et al. 2014) winds events from the southeast, and in part to the Yucatan current that floods the Yucatan Shelf from the east (Abascal et al., 2003). However, during CACS events, both (i) winds (from Northwest and North) produce a larger and northerly shear stress on the sea surface, and (ii) pressure gradients due to atmospheric pressure perturbations, drive water towards the Peninsula from the north and northwest. As a consequence, the predominant longshore current switches direction from northwest to southeast and leads to an increase of the sea level along the Northern Yucatan coast and hence inside the coastal lagoons. It is evident from Figure 13 (panels a and b) that the storm surge during Event A (the event with the longest duration) was larger than Event B both outside and inside the lagoon, even if the peak wind speed for each event was similar (see Figure 4), suggesting that the duration of the storm is an important factor for the generation of storm surges. It can also be seen from these two panels that the surge is greater in the eastern portion of the lagoon due to the wind stress orientation – which accounts for a significant amount of the surge – and the corresponding wind set-up, in addition to the water volume inflow through the inlet.

Regarding the wave set-up (panels c and d), Event A presented larger values (0.35 m along the Progreso coast and roughly 0.17 m inside the Chelem lagoon) compared to Event B (0.3 and 0.14m, respectively). These differences seem to be related to the tidal phase and wave off energy (Dodet et al., 2013) as well as to the wave direction (Guza and Feddersen, 2012) when the storm peak took place in each event.



In order to further investigate the evolution of the wave set-up through the inlet during the two TSSE peaks during Event A (times t1 and t2 in Figure 5), *Figure 14* shows the profiles of TSSE, storm surge, wave set-up, the cross-shore velocity (V), and the wave height in the transect perpendicular to the coast showed in Figure 13 (panel c). The 9 km long transect passes through the inlet and starts 1,000 m inside the lagoon (i.e., the coastline is at x=1,000m). For time t1 the TSSE (top panel, black continuous line) was

higher in the sea side than in the inlet channel lagoon. Then, the sea level slope for that moment induced flood currents at the inlet (see V total for t1 in bottom panel, left axis). The TSSE was lower than the storm surge height because the astronomic tide level was negative for t1 and t2 (Figure 5), being t1 the time when the TSSE reached its maximum value for Event A. The maximum wave height for t1 (bottom panel, right axis) before the breaking point was higher than 2.8m, but due to the shallowness of the Yucatan platform, waves experienced a gradual but significant dissipation along their propagation toward the coast. The peak wave

period and the azimuthal mean wave direction for t1 were 6s and 308° (49° respect to the coast, from northwest), respectively (not shown). The wave set-up reached 0.08m inside the main channel and 0.07m at the breaking point.

For t2, the TSSE in the inlet channel was higher than offshore (12 cm difference), inducing an ebb flow (see V total for t2 in bottom panel, left axis). Besides, the storm surge dropped significantly (more than 40 cm offshore) because it corresponded to the end of the storm, with weaker winds, and its effects on the storm surge where receding. In terms of the wave climate, the maximum wave

height for t2 decreased with respect to t1 (from 2.8 to 2.4 m), the peak wave period increased from 6 to 11s – suggesting that longer swell waves reached the coast by the end of the storm –, and the mean wave direction became more cross-shore (76° compared to 49° at t1). Under these conditions, the wave set-up reached 0.17m in the inlet channel and 0.14m at the breaking point, values significantly larger that at the peak of the storm (t1). In fact, the wave set-up at the inlet channel represented 7% of the wave height at breaking point for t2, and only 2.85% for t1. However, for the Progreso beach, where wave set-up reached 0.35m (see Figure

13, bottom, left panel), it was the 14.5% of the wave height at the breaking point.

From the analysis described above it is seen that the maximum wave set-up for Event A occurred at time t2 when the mean wave direction reached its maximum northerly value and when the ebb reached its maximum. Indeed, the wave set-up is controlled by the $S_{xx}$ radiation stress component and reaches its maximum when the incident wave direction is normal to the coast(Guza and Feddersen, 2012).

**4.4 Flooding prone areas for events A and B.**

Regarding flooding from events A and B (Figure 15), the most affected area in the sea side seems to be the stretch of coast from the inlet to the Progreso Pier, partly due to its concave shape. Flooding on this area was larger during Event B (bottom panel) because this CACS event hit the Peninsula during high tide and the TSSE was larger. Inside the lagoon, the flooding is in general larger than in the open coast for both events, due to its small capacity to regulate increased volumes of water flooding through the

inlet as well as the wind set-up that can be larger for some wind directions than in the open coast. In fact, even if the sea surface elevation in the seaside is larger for Event B than for Event A (as shown in section 3.3), higher water level inside the lagoon, particularly in its eastern sector, was found during Event A due to stronger wind set-up contribution as mentioned above, as well as a stronger wave set-up, resulting in larger flooded areas of Progreso. In terms of the specific flooding areas in the town of Progreso (east side of the inlet), the total number of blocks affected during Event A was 157 (25 % of the town surface area), where

8 of them are located along the Progreso beach and 149 in the eastern lagoon shores. For Event B, the total number of blocks flooded was 110 (18% of the town surface area; 18 on the sea side and 92 on the lagoon side). These results show that the most flood prone areas at Progreso are the ones located in the southern area, bordering the Chelem lagoon, and not the coast, mainly because of the semi-enclosed nature of the lagoon and the wind and wave set-up that occurs during storms.





**5 Discussion**

We investigated the role of tidal modulation on the CACS events by means of numerical model. Since (i) Event A flooded larger areas than Event B, even if it occurred during low tide, and (ii) Event A could have occurred during other periods of the tide, a numerical experiment was carried out to assess the tidal phase effect in the flooding, the wave set-up and the residual sea level

induced by CASC over Progreso (using the four tide scenarios shown in Figure 5).

**5.1 Flooding at Progreso modulated by the tidal phase and the wind set-up inside the lagoon**

Figure 16 shows the maps of maximum flood (TSSE) corresponding the the four tide scenarios done for Event A (reminding that TS3 corresponds to the actual tidal phase that occurred during Event A). It is important to note that the time at which the maximum flood ocurred is not the same in the sea side and inside the lagoon, and is not the same for each simulation, similar to the criteria

used with flood risk analysis in other studies (Zachry et al., 2015). It can be seen that the flood worst case scenarios were for TS1 (high tide during the peak of the storm) and TS4 (rising tide near mean sea level during the peak of the storm), and significantly lower with tide scenarios TS2 and TS3 (see summary in Table 1). This is partly due to the fact that during the storm – between 00:00 and 15:00 of March 13, 1993 (see Figure 5, lower panel), i.e., during the period where the local winds where stronger and able to produce large wind set-up inside the lagoon –, the tide was high for TS1 and TS4 and low for TS2 and TS3. Therefore, the

wind and astronomic tide effects added up for TS1 and TS4. In turn, for TS2 and TS3, the tide was low and the dominant factors were the residual sea level and the wave set-up (Figure 17, top panel). Comparing TS1 and TS4, it can be seen that the highest sea side TSSE occurs with TS4, due to the eastward tidal currents during rising tide that contribute to the storm surge (not shown), while for TS1 the tidal currents were westward during receding tide and did not contribute to the storm surge. However, inside the lagoon, the maximum flood for TS1 occurred during stronger local winds (07:00) that generated a larger wind set-up, while for

TS4 it occurred 4 hours later, when the local wind intensity was receding and thus producing a smaller wind set-up. Therefore, when the wind direction is parallel to the main lagoon axis, a large wind set-up is generated at the eastern part of the lagoon. As for the results for TS2 compared to TS3, the flood levels occurred at the same time (07:00) but were higher for TS2, in particular inside the lagoon. This is mainly due to the fact for TS2 the tide was receding and for TS3 the tide was in slack-low, which in turn produced a larger wave set-up inside the lagoon for TS2 (Figure 17, top panel, right axis). This is in agreement with what described

(Dodet et al., 2013), where they stated that during ebb (receding tide), waves break over the ebb shoal, leading to stronger values of wave radiation stress term than during flood (see wave set-up for TS1 in Figure 17), resulting in larger wave set-up that propagates inside the lagoon. In summary, the astronomic tidal phase during the passage of the storm is a very important factor for flooding, not only because of the tidal level itself, but because of the interactions with the other contributors to the TSSE. For instance, Figure 16 shows the city areas (blocks) affected for the different tide scenarios, showing that a storm with the

characteristics of Event A would have been much more destructive if it had occurred during high tide, as in TS1 and TS4.

**5.2 Residual sea level and wave set-up modulated by the tidal phase.**

Figure 17 shows the residual sea level (top panel, left axis) and wave set-up (top panel, right axis) for Event A under different tidal phases as well as the wind speed offshore (middle panel, left axis) and the significant wave height (Hs, middle panel, right axis) used as forcing for the numerical experiment. In the bottom panel are shown the TSSE (left axis) and the cross-shore velocity V.

The residual sea level was obtained by means of a harmonic analysis using the T_Tide program (Pawlowicz et al., 2002), from which the residual tide was determined by means of the sea surface elevation for each scenario at the Chelem inlet from the HD model using as boundary conditions the Configuration 4 described in section 3.3. From these numerical experiments, it was found



that for this site the residual tide is larger during low (TS3) or receding tide (TS2), while it is smaller during high tide (TS1) or rising tide (TS4), and the variation is nonlinear, as found in other studies (Lin et al., 2012; Rego and Li, 2010), who attributed this behavior to nonlinear effects of the bottom friction and momentum advection on the surge due to the presence of the tide. This is also confirmed by Horsburgh and Wilson (2007), who, in a study for the coast of Great Britain, stated that surge peaks never occur

during high water.

As already mentioned in section 3.3, while the local wind contributes to the wind set-up inside the Chelem lagoon, the offshore wind (e.g., at B location, see Figure 3) has a good correlation with the residual sea level. For instance, the second peak in the offshore wind speed (location B) is also present as a second peak in all the residual sea level scenarios as well as in Hs.

On the other hand, model results from this study show that the wave set-up contribution inside the Chelem lagoon is tidally

modulated as found in other studies for other sites (Dodet et al., 2013; Olabarrieta et al., 2011; Smith et al., 2000; Smith and Smith, 2001). *Figure 17* suggests that the maximum wave set-up occurred during ebb (positive values for V-velocity component), and large values of wave set-up are maintained as long as the ebb stays strong, as can be seen for TS4 (blue solid lines in top a lower panels). Dodet et al. (2013) observed similar inlet hydrodynamic behavior from data analysis and numerical modeling at the Albufeira lagoon (in Portugal). They stated that during ebb, currents cancel the intrinsic group velocity at the inlet, and waves are

refracted by the ebb-jet current at the entrance of the inlet. Furthermore, this ebb-jet current induced the increase of the wave height at the inlet, leading to increase the energetic breaking, and thus more wave set-up contribution inside coastal lagoons. Similarly, Gonzalez et al. (1985), who observed from a case study at the Columbia River entrance that wave height increases during ebb and decrease during flood. The opposing current retards the advance of a wave and even block the wave energy transport when the upstream component of the wave group velocity is equal to the current velocity, and the flood-induced current enhance the advance

of the wave (Olabarrieta et al., 2011) but does not contribute significantly to the wave set-up.

Besides, Olabarrieta et al. (2011), who identified the effects of wave-current interaction on the circulation of the Willapa Bay (Washington State), showed that the wave set-up inside estuaries increase with the offshore energy, and Malhadas et al. (2009) suggested that not only the offshore significant wave height, but also the inlet morphology (mainly depth and length), lead to induce more or less wave set-up inside estuaries. They stated by means of numerical solutions of simple idealized models that the

shallower and longer the morphology, the more the wave set-up is increased. This is in fact the case in this study, where the Chelem inlet is 130m wide, and the inlet channel is roughly 1.2 km long and 3m deep (decreasing further along the channel), and it was found that an important wave set-up was induced inside the lagoon during storm events.

From the above, it is seen that the maximum residual tide and the maximum wave set-up did not occur at the same time at the inlet for any of the tidal phase scenarios used, but it does not seem to be determinant in the high flooding levels. Instead, what seems to

play a more important role is the duration of these maximum values of residual tide and wave set-up. In fact, the highest values of TSSE at the inlet, and occurring during a longer time, are found with the TS4 tidal phase scenario. This is mainly due to (a) a longer duration of significant residual tide and wave set-up, and to (b) a rising astronomic tide, which translates in higher astronomic tidal levels and is associated with easterly tidal currents that contribute to the sea level anomaly.

From this study, it was found that the most important contributions in order of significance during CACS pass over the Peninsula

are: the residual tide, the astronomic tidal phase, the wind and wave set-up (inside Chelem), and finally the Yucatan aquifer discharge which was found to be negligible.

## 5.3 Flood return periods

Regarding the nonlinearity in the processes contributing to flooding and the return periods, the extreme sea level analysis showed that the astronomic tide has an important effect to determine return periods for possible sea levels. In fact, the CACS events can



occur at any tidal phase and thus the CACS residual tide is, from the occurrence point of view, independent of any simultaneously occurring tidal phase. However, the interaction between residual and astronomic tides is an interesting subtle point to study: if this interaction is linear, the storm tide (TSSE without waves) probability would be roughly equal for each tidal phase, and would be simply equal to the sum of the residual sea level and the astronomic tide, which means independence between the astronomic tide

and the residual sea level. In that case, the use of joint probability methods could be used. These methods provide the chance of source variables taking values at the same time, and creating a scenario where a flooding event may occur. This method is usually used for independent events (Chini and Stansby, 2012). For instance, Zhong et al. (2013) assumed independence between the astronomical tide and residual tide and estimated the storm tide probability of a joint probability method. However, hydrodynamic numerical experiments for Progreso in the present study suggested that the astronomic tide and residual sea level have a nonlinear

relationship, and thus applying the joint probability method for this case would not be adequate. A pragmatic and simplistic approach can consist of using the a large data set of sea level reanalysis (for instance thirty years in this case), assume that a sufficiently large number of combinations of storm surges and astronomic tidal levels are present in the data set, and perform an extreme analysis of the sea level as a single variable. However, if quantifying the nonlinear effect would be the scope of any future study for Progreso, there are other options for estimating the storm tide probability given a height of the astronomic tide and phase

when the storm surge arrives (assuming that the surge can happen at any time during a tidal cycle with equal likelihood) such as the one proposed by Lin et al. (2012). They developed an empirical function based on over 200 extreme tropical cyclones event (with both, the storm tide and the storm surge simulated for the full range of tidal phases) for New York city. The implementation of this method for this zone is beyond the aim of this study, being the main objective for this research to study the hydrodynamic and flooding prone surfaces areas for Progreso during CACS pass. However, this study shows the necessity to develop an empirical

probability function based on the data for this area to estimate the probability of any storm surge given a phase of the astronomic tide as well as the wind direction and intensity with respect to the lagoon main axis.

## 6 Conclusions

This study has performed a thirty-year sea-level hidcast using a hydrodynamic model forced by tides, mesoscale currents, as well as wind and pressure fields from the CFSR. This hindcast information allowed to identify extreme water levels using the GEV

distribution function. Furthermore, the role of wave setup was also investigated for two selected storm events which correspond to the largest residual tide (Event A) and to the largest storm tide (Event B).

Regarding the four diferent boundary configurations used for events A and B, it was found that (a) the wave set-up and storm surge in Event A were large due to stronger wind instensity and longer duration than for Event B, and Event A occurred during ebb where the wave breaking at the inlet is stronger than during flood; (b) in terms of TSSE, Event B was higher in the sea side at

Progreso because it ocurred during high tide, but in the lagoon side the TSSE was higher for Event A because of the wind stress over the water surface. Since the most populated area at Progreso is the back-barrier lagoon shore, Event A caused more flooding. Inside the lagoon the local winds play an important role, especially when the wind direction is parallel to the main axis of the basin, producing large wind set-up. In terms of flooding, the most affected areas for both storm events (A and B) occurred in the eastern shores of the lagoon, due to significant wind and wave set-up inside the lagoon, in particular for Event A. In this sense, the sea

surface elevation inside Chelem during CACS pass is not only important because of the exchange water through the inlet, but also because of the role of the wind and wave set-up over the lagoon, as well as nonlinear interactions between these forcing agents and the astronomic tide.





Based on results from events A and B, we investigated the role of the tidal phase on the residual sea level, the wave set-up and the total flood area in Progreso. Furthermore, numerical experiments varying the tidal phase for Event A was carried out. The results suggest that both, the wave set-up and residual sea level are tidally modulated: the maximum wave set-up inside the Chelem lagoon occurred during receding tide (ebb) when the ocean water level was near mean sea level and the incident wave direction was almost

is perpendicular to the coast. The residual sea level was larger during low or receding tide, and smaller during high or rising tide. However, the maximum flooding surface areas over Progreso occur when the CACS peak coincides with rising tide near zero level or high tide (TS4 and TS1 scenarios). Nevertheless, the maximums for the residual sea level and the wave set-up did not occur at the same time in any of our numerical experiments. The tidal phase difference with respect to the storm arrival determined the flood duration and the maximum water depth reached for each scenario.

If the largest CACS residual sea level (Event A) had occurred during high spring tide instead of low tide, the percentage of blocks flooded in the city would have increased from 25% to 60%. The latter implies the need to estimate accurately the probabilities of residual and tidal levels, in conjunction to local winds and wave setup for a reliable estimation of coastal flood hazard caused by CACS events. This requires the definition of empirical probability functions specific for the area based on the astronomic tidal amplitude and phase, storm surge, wind set-up (inside the lagoon) and wave set-up.

**Acknowledgments.** W.R. was supported with a doctoral scholarship with CVU 308087 from the Mexican National Council for Science and Technology (CONACYT), complemented with support from the Council of the National Research Training (COLCIENCIAS), the DGAPA of the National Autonomous University of Mexico (project IN111916), from CONACYT the National Coastal Resilience Laboratory (project 271544), and project INFR-2014-01-225561, and project 5341 from the

Engineering Institute at UNAM. W. R. thanks DHI Water & Environment for facilitating a student license of Mike 21 SW and HD, Dr. Cecilia Enriquez for providing the bathymetry of the Chelem lagoon, and also Pablo Ruiz-Salcines from UNAM and Jaime Hernandez-Lasheras from Cantabria University for their help and suggestions to this study.

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




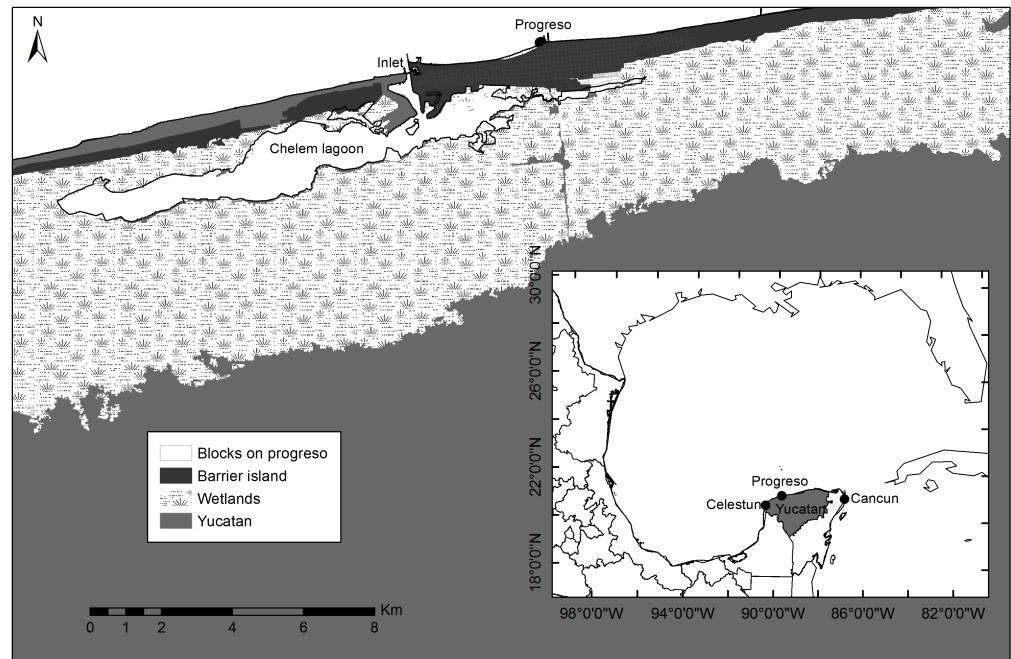

**Figure 1. Location map indicating the study zone and the town of Progreso**

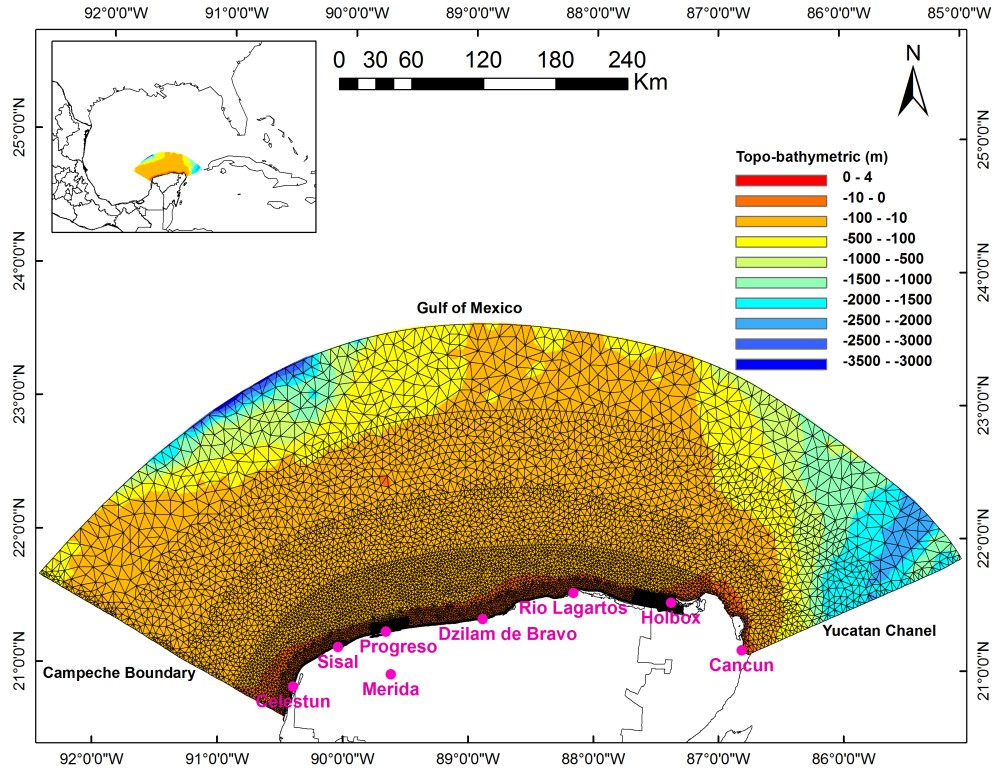

**Figure 2. Computational domain for both hydrodynamic and coupled model (Mesh 1)**




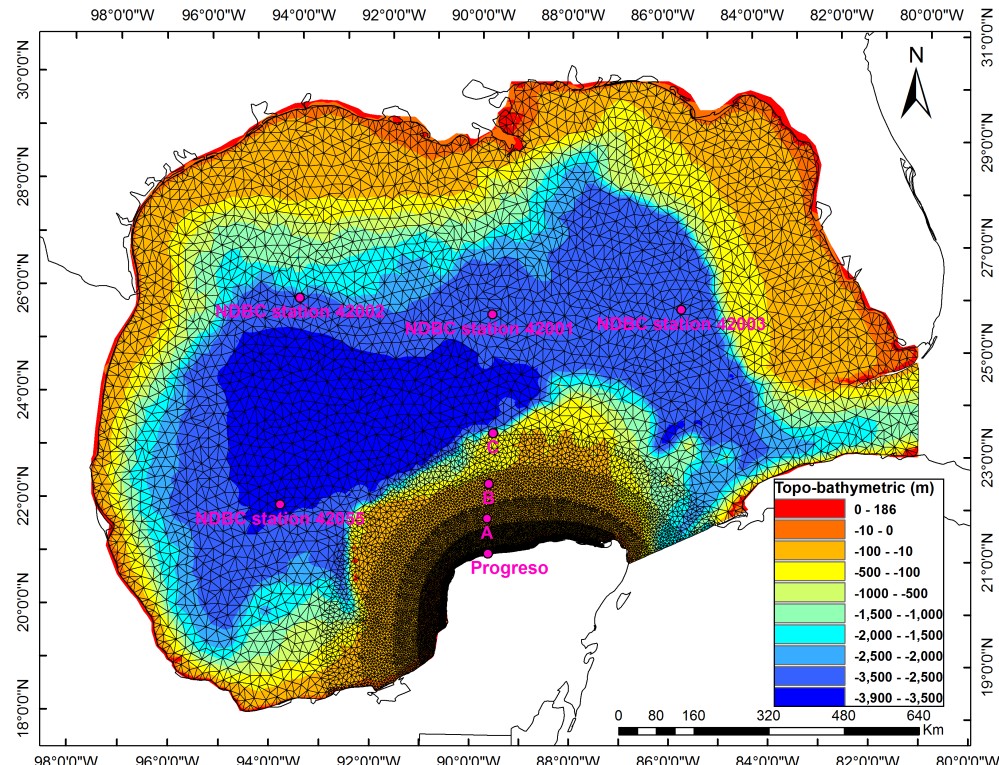

**Figure 3. Wave model computational domain and topo-bathymetric for the GoM (Mesh2), and location of NDBC stations used in this study.**

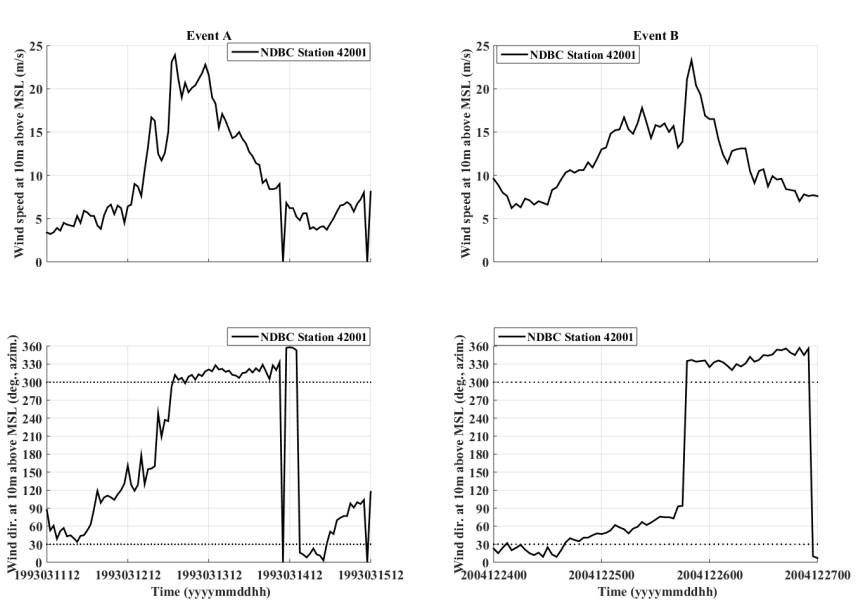

**Figure 4 NDBC station 42001 data: Wind speed (10m above sea level) and direction for events A (left column) and B (right column).**



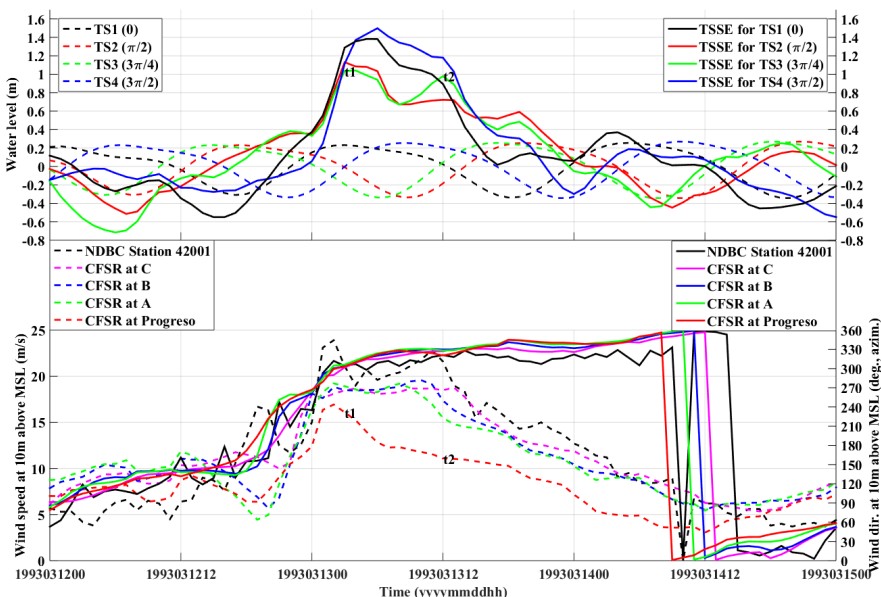

**Figure 5. Top panel: tide forcing for the four scenarios at Progreso, for Event A (dashed lines); time series of TSSE at Progreso (continuous lines), which will be discussed in the results section. Bottom panel: wind speed (dashed lines) and direction (continuous lines) at the five locations shown in Figure 3.**

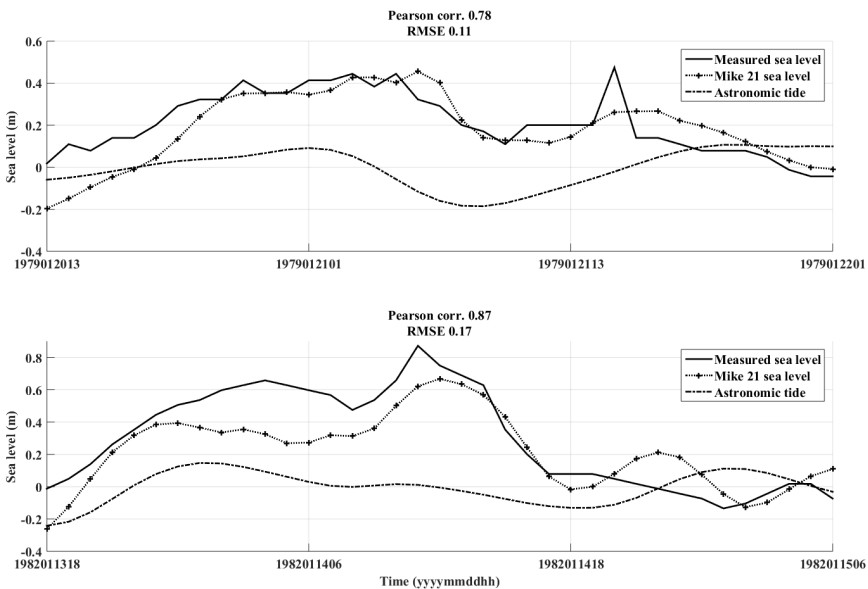

**Figure 6. Hydrodynamic model validation with tide gauge data from Progreso port in Yucatan. Top panel: validation for a CACS event in 1979. Bottom panel: validation for a CACS event in 1982.**





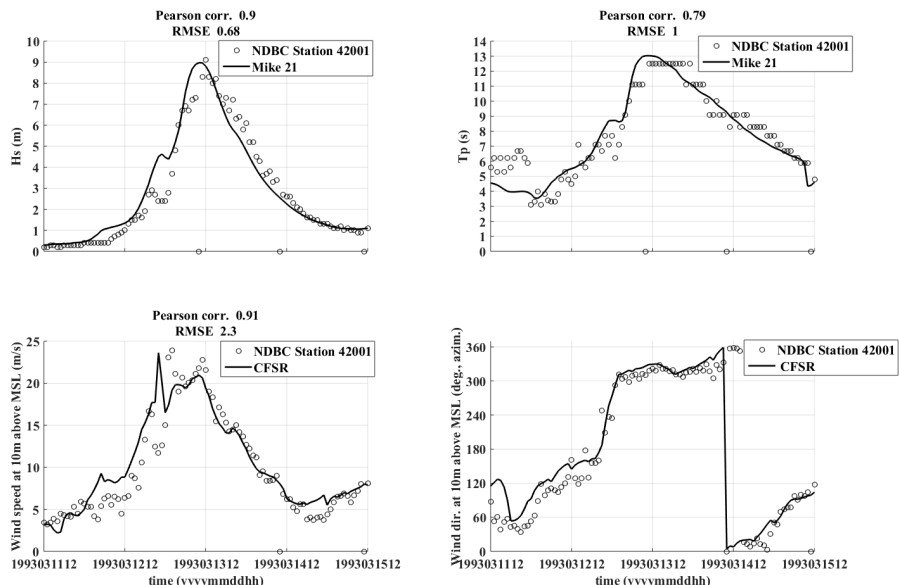

**Figure 7. Wave model and CFSR wind validation for Event A. (Left top panel) Significant wave height. (Right top panel) Wave peak period. (Left bottom panel) Wind speed 10 m.  (Right bottom panel) wind direction.**

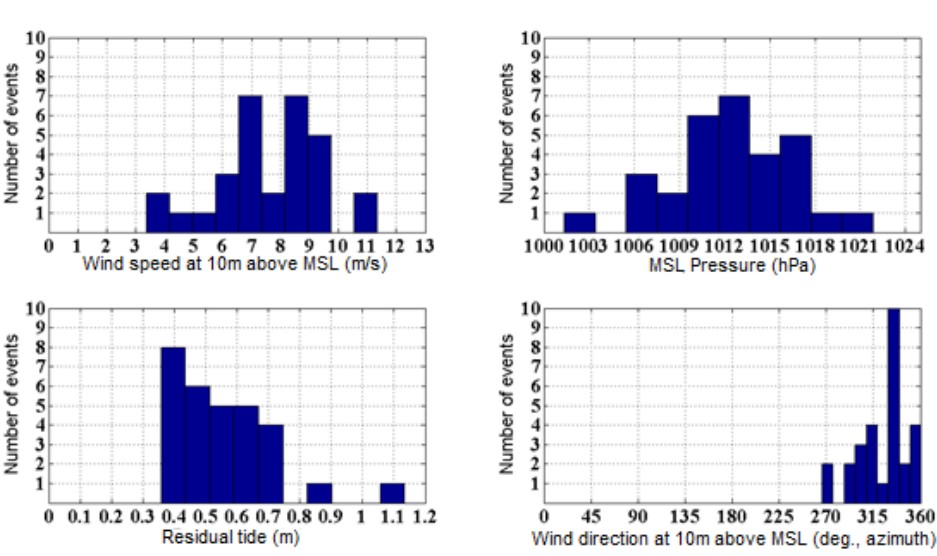

**Figure 8. Characteristics parameters of the 30 events associated with the annual maximum residual tide at Progreso.**

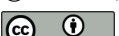


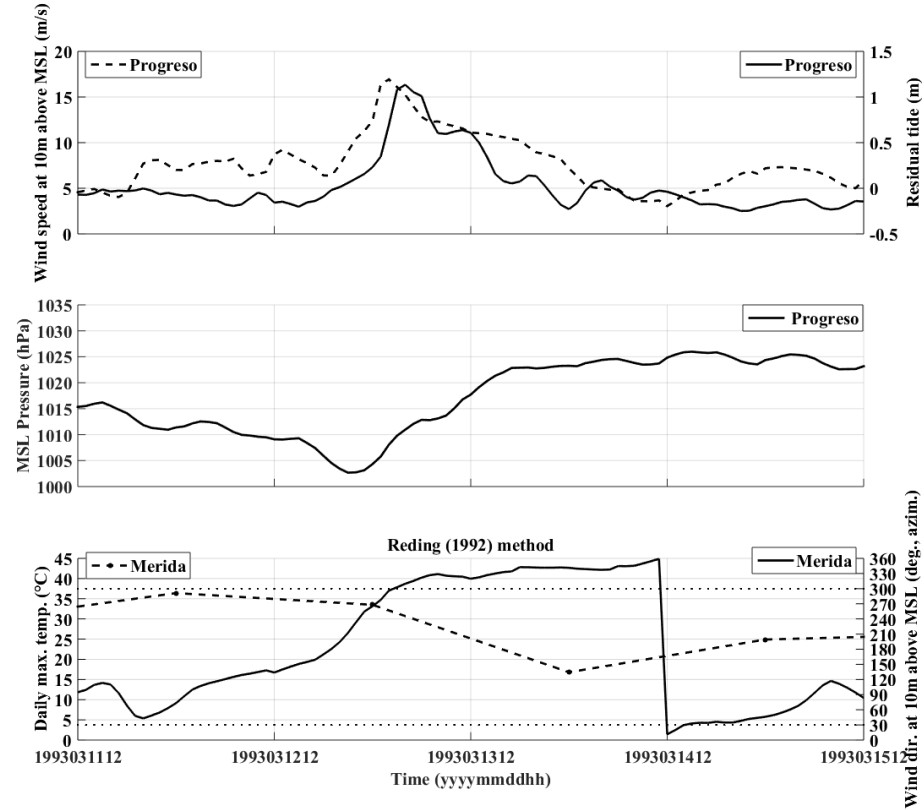

**Figure 9. Characteristic parameters for the CACS event that induced the highest residual tide in Progreso (Event A). Top panel: Wind speed and residual tide. Middle panel: Mean Sea Level pressure in Progreso. Bottom panel: daily maximum temperature and wind direction. The dotted lines in the lower panel correspond to the 30-300º range of wind direction associated with CACS.**





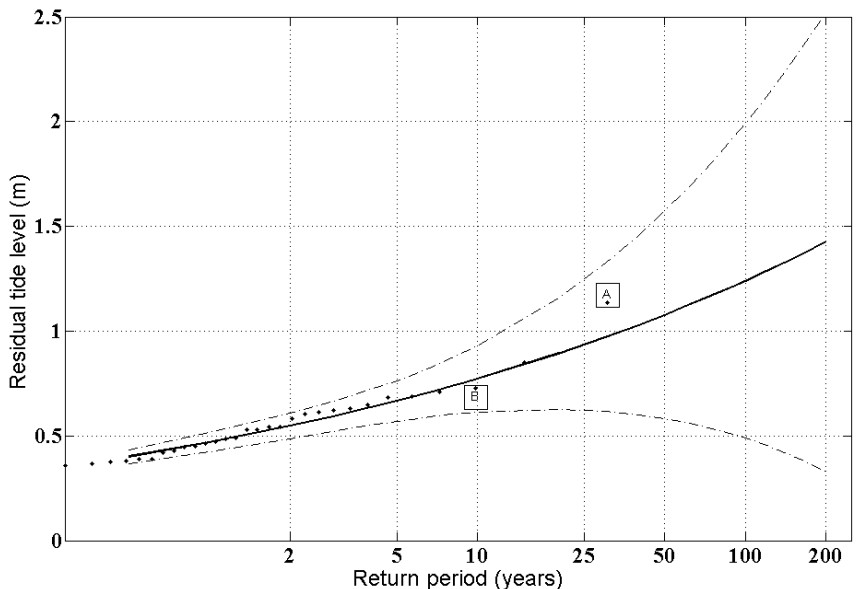

**Figure 10. Yearly maximum residual tide heights adjusted to the GEV distribution probability function for Progreso, using dataset D1 from section 3.3. The solid curve is the fitted residual tide; the dots are the 30 annual maximum residual tide values from the hydrodynamic numerical simulation, and the dashed curves are the 95% confidence limits.**

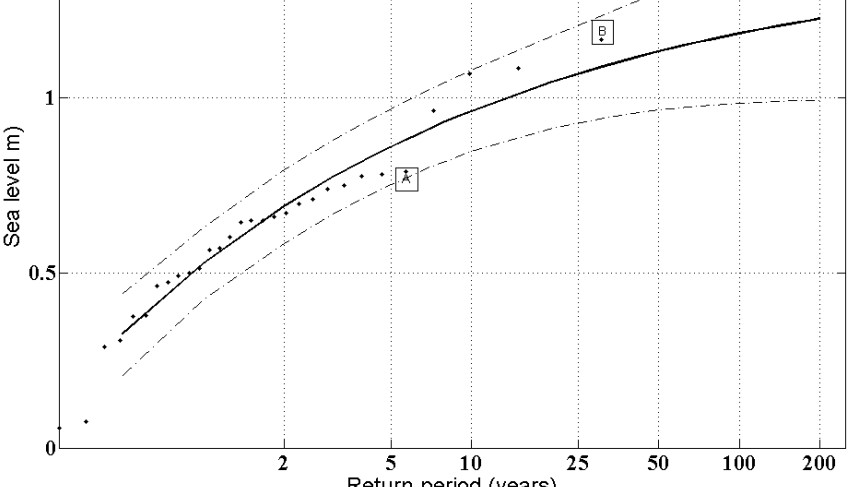

**Figure 11. Yearly maximum sea level height (using model Configuration 4) adjusted to GEV distribution for Progreso, using dataset D2 from section 3.3. The solid curve is the fitted sea level height; the dots are 30 annual sea level heights associated to annual maximum residual tide values from the hydrodynamic numerical simulation, and dashed curves are the 95% confidence limits.**




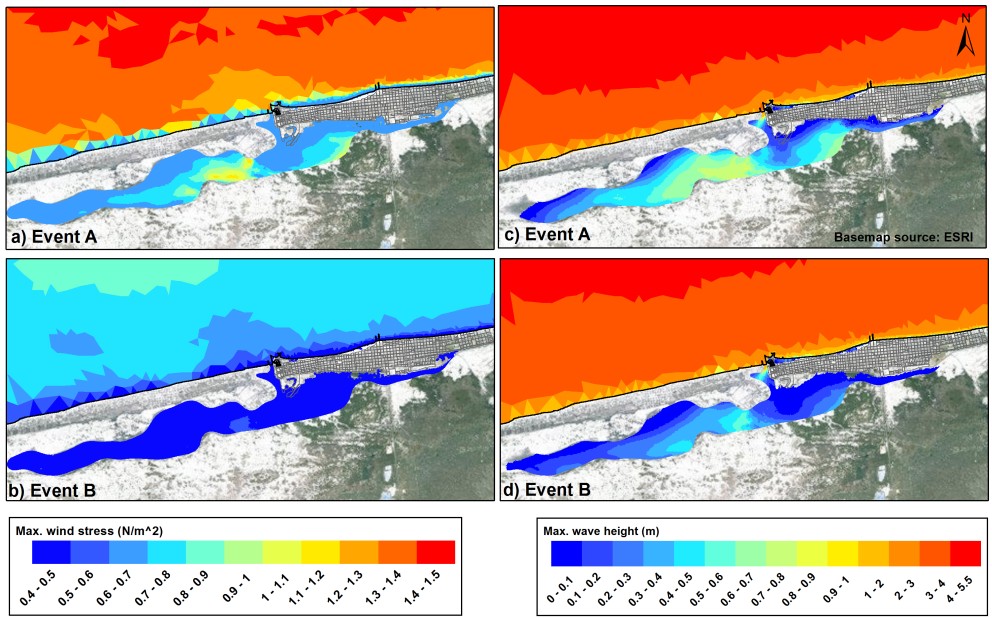

**Figure 12. Highest wind stress values during Event A (panel a) and Event B (panel b), and highest wave height values for Event A (c) and Event B (d).**

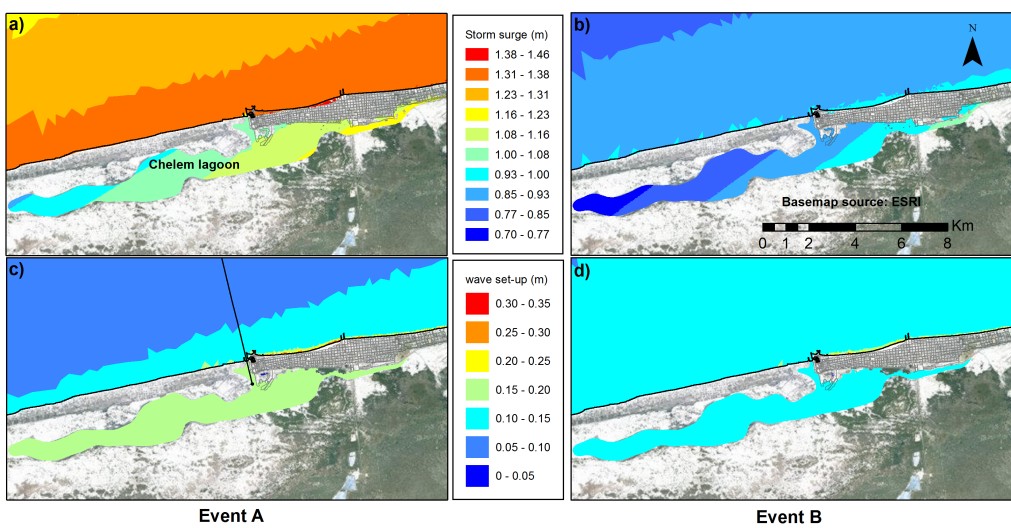

**Figure 13. Main contributions for maximum flooding for Progreso during Events A and B: (a) Maximum storm surge for Event A. (b) Maximum storm surge for Event B. (c) Maximum wave set-up for Event A. (d) Maximum wave set-up for Event B**




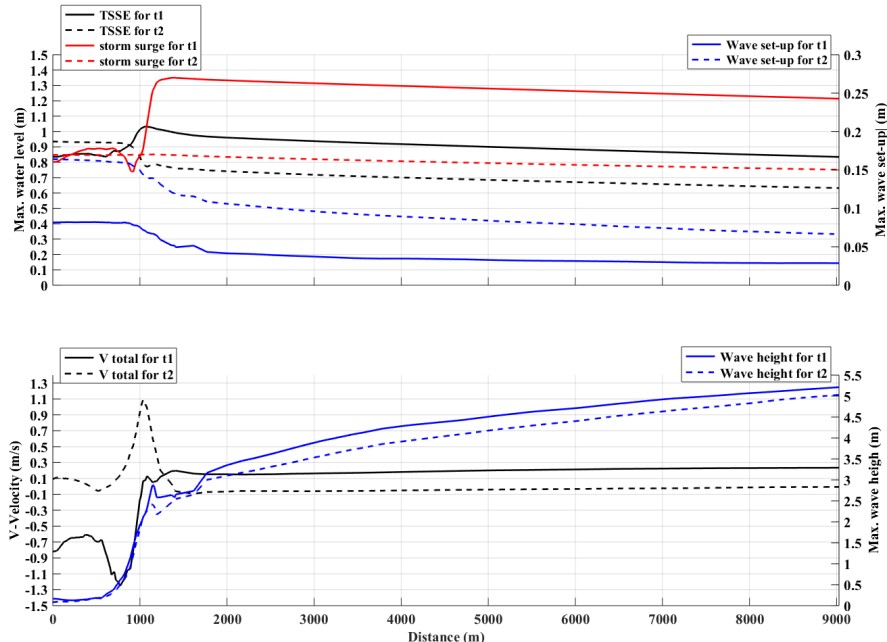

**Figure 14. Modulation of the wave set-up for the ebb/flood current at the Chelem inlet. (Top panel) TSSE, storm surge and wave set-up for t1 and t2. (Bottom panel) V Total component velocities and maximum wave height for t1 and t2.**

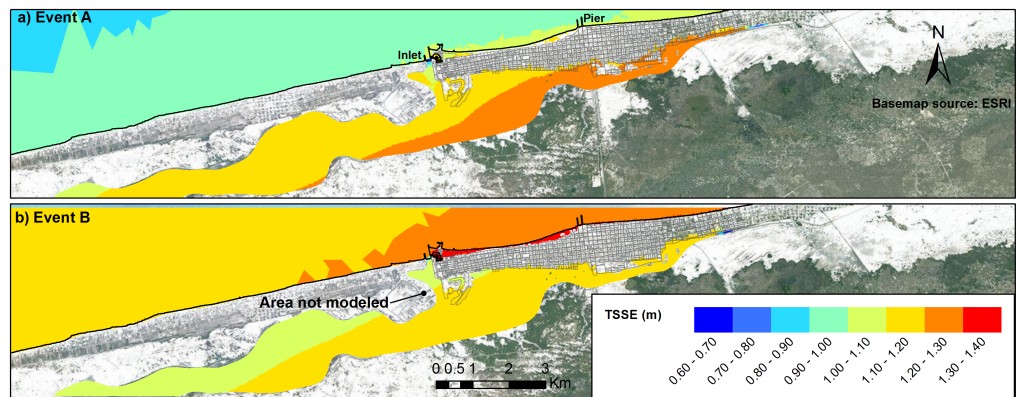

**Figure 15. Maximum flooding over Progreso during events A and B. (a) Flooding areas for Event A. (b) Flooding areas for Event B**



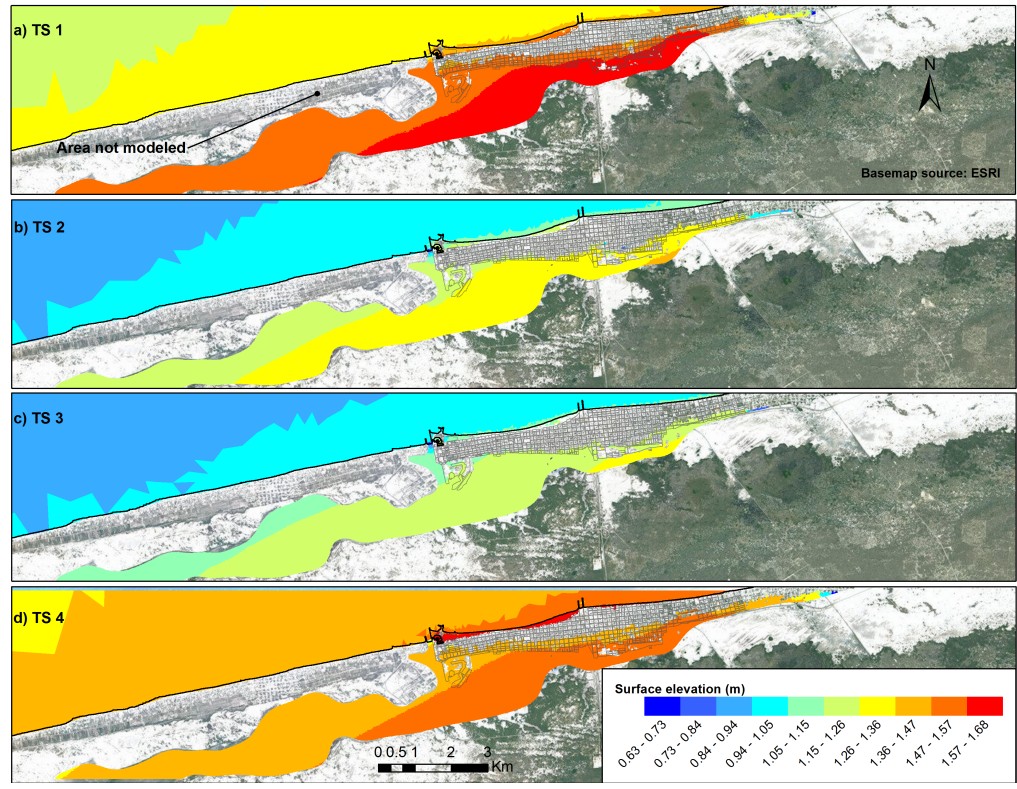

**Figure 16. Flooding maps with Total Sea Surface Elevation at Progreso under four tide scenarios for Event A: a) Maximum flooding for TS1. b) Maximum flooding for TS2. c) Maximum flooding for TS3. d) Maximum flooding for TS4.**





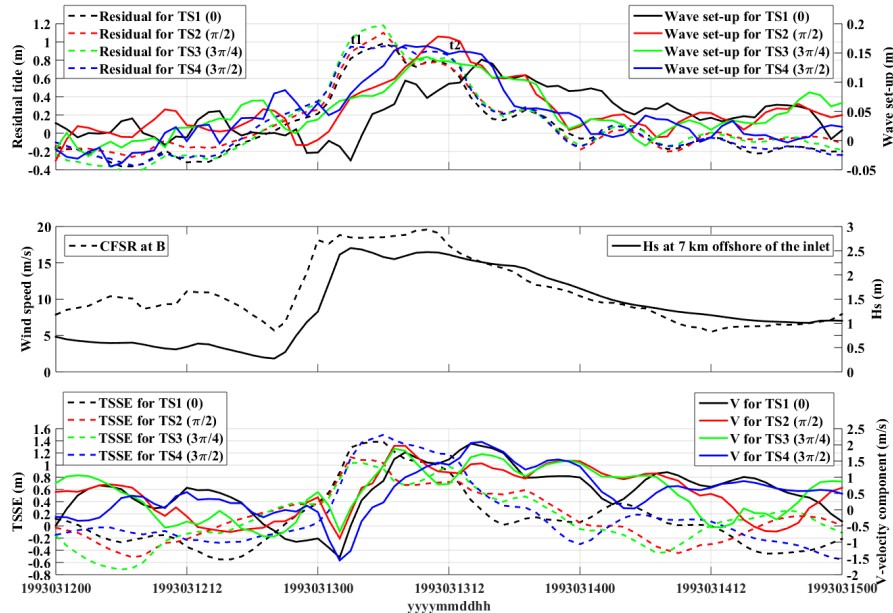

**Figure 17.Residual sea level and wave set-up modulated by the tide. Top panel: Residual sea level and wave set-up for the fourth tide scenarios. Middle panel: Wind speed at 10m above MSL and Hs used as forcing for the numerical experiment. Bottom panel: Total Sea Surface Elevation (from the HW model), TSSE and the total V-velocity component (from the HW model) perpendicular to the inlet.**

**Table 1. Blocks of Progreso affected as a function of tidal phase during Event A.**

| Tide Scenario | Total blocks affected | Percentage of total blocks | Sea side blocks affected | Lagoon side blocks affected |
|---|---|---|---|---|
| TS1 | 368 | 60% | 19 | 349 |
| TS2 | 199 | 30% | 12 | 187 |
| TS3 (Event A) | 157 | 25% | 8 | 149 |
| TS4 | 354 | 57% | 33 | 321 |

