# Peer review of "Assessment of coastal flooding and associated hydrodynamic processes on the Southeast coast of Mexico, during Central American Cold Surge events"

_Natural Hazards and Earth System Sciences, 2017_

## Referee Comment (RC1) · Anonymous Referee #1 · 18 May 2017

The paper deals with the assessment of coastal flooding in the Yucatan Peninsula (Mexico) and, particularly, within the Chelem lagoon, near the city of Progreso. The paper is an interesting case study, which reports an in-depth numerical analysis of the coastal dynamics that lead to coastal flooding in a given location.

The paper is quite long and not well organized. Mainly for this reason, and for the inadequate English, a substantial improvement is required to meet the standards of the Journal.

Major points

1) The English of the paper needs to be thoroughly revised. Many typos, awkward and

incorrect sentences can be found in the text and should be fixed. The help of a native speaker could be beneficial.

2) The paper actually deals with coastal flooding in a specific location. It does not assess, in general terms, the role tidal modulation in coastal flooding. Please consider to revise the title, as tidal modulation is only one aspect of the overall process analyzed in the paper. I'm thinking of something as "Assessment of coastal flooding in the Yucatan coast during Central American Cold Surge Events".

3) The presentation of the paper is very confused. Different type of data, validation of model results and of input data (i.e., water residuals and wind data), characterization of CACS events and analyses of the entire 30-tears time series, etc. are mixed together. The Authors should re-order, and possibly shorten, the paper, which needs to be less dispersive to the reader attention. Rather than being a vast report of all the analyses carried out by the Authors, a scientific paper should lead the reader to clear conclusions.

4) The Introduction is too long; many specific information should be moved from Introduction to Section 2.

5) Although an interest topic per se, I do not understand the role of hydrogeology in coastal flooding. How can an aquifer discharge affect the sea level? This question needs to be clearly assessed (if aquifer discharge actually plays some roles), or otherwise, being not even mentioned in the paper.

6) The "References" section contains many typos and missing/wrong information (the formatting of conference proceedings and book chapter has problems with the conference/book title and with the number of pages). Please check carefully all the details of each bibliographic item.

Minor points

-Check for the presence of double consecutive spaces into text.
-The use of acronyms and abbreviations should be limited, as it makes difficult to follow the text for those readers that are not already familiar with them. Finally, make sure that all the abbreviations are properly introduced when they first appear.

-Abstract: the reason why hindcast sea level time series was used (i.e., the lack of measurements) has to be stated. Rather than reporting specific numerical data, please specify the locations object of the study (Progreso and Chelem lagoon) and clearly outline the analyses carried out and the main results.

-page 1, line 11: an "... occurrence probability" can not be performed.

-p. 1, l. 16: "inlet" of what?

-p. 1, I. 17: "despite micro-tidal conditions" what does this means? What is the difference between the tide (mentioned just before) and these "micro-tidal conditions"?

-p. 1, l. 34: "passing over the GoM"

-p. 2, l. 4: "methodology" pertains to the "study of methods", use "method" instead. (see also p.6, l. 14)

-p. 2, l. 14: replace "induced by ... on the sea surface and" with "enhanced by"

-p. 2, l. 15: Shorten the sentence as "Consider that the effect of pressure field is relatively small during high-pressure atmospheric systems as CACS (Flather, 2001)."

-p. 2, l. 19: which currents?

- -p. 2, I. 22: "... flood hazard is defined..."
- -p. 2, l. 23: "... and period; it depends on..."
- -p. 2, l. 24: delete "However"
- -p. 2, I. 25: The year is missing in the reference to Dorrestein
- -p. 3, l. 13: delete "However"

NHESSD
-p. 3, l. 28: "back-barrier lagoon of Chelem, behind Progreso"

-p. 3, l. 32: Start a new paragraph with "In terms of hydrogeology..." (see also major point n. 5)

-p. 3, l. 34: delete "in"

-p. 4, l. 17: units: remove periods from within units' expressions (e.g., m3s instead of m3.s). Please check throughout the text

- -p. 4, l. 17: put a reference to Fig. 2 after "Holbox".
- -p. 4, l. 22: please define "HD"

-p. 4, l. 23: "shallow water equations", not "shallow waters equations"

-p. 4, l. 25-26: awkward sentence

-Eq. (1), (2), and (3): What kind of discharge is S, whose units are 1/s? How are the components of the "lateral stress" evaluated? Are these Reynold/dispersion stresses?

-p. 5, l. 4: delete "studying". This sentence seems incomplete.

-p. 5, l. 10: After "wave action equation" please put a reference (bibliographic or to an equation reported in the paper).

-eq. (4) is correctly written?

- -p. 5, l. 15: the sentence ": the directionally ... formulation" is duplicated
- -p. 5, l. 18: "as described in"
- -p. 5, l. 21: what's the meaning of "(10-10 km)"?
- -p. 5, l. 22: "and both swell and combined... are not important"
- -p. 5, l. 27: I don't see S in the equation
- -p. 6, l. 9: "as reported"
-p. 6, l. 14: Does the last sentence refer to the previously described treatment of the boundary condition? In this case, this sentence should be moved before the description of the boundary condition.

-p. 6, l. 19: "according to Arcement and Schneider (1989)", "of the Yucatan sand"

-p. 6, l. 22: what is the result of the further calibration of Cd?

-p. 6, l. 37-ff: This paragraph should be reorganized. The risk of coastal flooding is only associated with the total sea level, not directly with the sea residual. Clearly, the analysis of the sea residual is crucial, e.g. in order to improve sea level forecasts, since the sea residual is affected by major uncertainties than the astronomical tide (e.g., Met et al., 2014).

-p. 7, l. 5-10: D1 and D2 are datasets, i.e. sets of data, but they are described as actions/procedures ("consisted in identifying", "consisted in adding"). In D1 the astronomical tide is removed, in D2 is added again... Please make the description of the two datasets clearer.

-p. 6, l. 13: datasets have to be denoted with D1 and D2, not with (a) and (b).

-p. 6, l. 16: "selected and then analyzed"

-p. 6, l. 26: "At the peak". Remove the comma after "were"

-p. 6, l. 31: "while" is a temporal expression, use "whereas for Event B..." instead. In addition, "closer to the normal to the coast".

-p. 8, l. 15-16: It is not clear to me how this goal was pursued. By shifting the astronomical tide for the entire 30-years time series?

-p. 8, l. 27-ff: As for what I understand, a hypothetical scenario (TSSE) is compared with a measured (reanalyzed) wind field. Does this make sense?

-p. 9, l. 27 Figure 10, not Figure 9.

NHESSD
-sect. 4 and sect. 5 are quite long. I suggest a sensibly shortening of these sections.

-p. 14, l. 11: "consist in using the... assuming that... and performing..."

-p. 14, l. 16: "events"

-p. 14, l. 23: A study can not perform anything...

-p. 14, l. 24: "to identify extreme water levels and characterize their probability of occurrence using..."

-p. 14, l. 27: "different". "conditions", not "configurations".

-p. 14, l. 31: The fact that an area is more populated can not be a cause of more flooding... Rather, it can cause greater damages...

-p. 14, l. 33: "producing large set-up"; and waves? See, .e.g., Carniello et al. (2005).

-p. 14, l. 33: "Chelem lagoon". "were" in place of "occurred".

-p. 14, l. 34-35: "The passage of CACS, besides affecting water exchange with the sea and renewal dynamics inside the Chelem Lagoon (Viero & Defina, 2016a,b), is show to produce significant wind and wave set-up, characterized by nonlinear interactions between meteorological forcings and the astronomic tide.

-p. 15, l. 1: "Based on modeling results from..."

- -p. 15, l. 2: "total flooded area"
- -p. 15, l. 5: delete "is"
- -p. 15, l. 6: awkward (and quite obvious) sentence.
- -p. 15, l. 14: "storm surge, and set-up due to both wind and wave".
- -Figure 3: I suggest putting the text in magenta on a white box to improve readability.
- -Figure 9, top panel: change the labels "Progreso" and "Progreso" with "Wind speed"
and "Residual tide".

-Figure 9, bottom panel: as for the top panel, labels should indicate the kind of data, not the location. "Reading (1992) method" is redundant here.

Additional references

Carniello L., Defina A., Fagherazzi S., D'Alpaos L., 2005. A combined wind wave-tidal model for the Venice Iagoon, Italy. Journal of Geophysical Research – Earth Surface, 110, F04007, doi:10.1029/2004JF000232.

Mel R., Viero D.P., Carniello L., Defina A., D'Alpaos L., 2014. Simplified methods for real-time prediction of storm surge uncertainty: The city of Venice case study, Advances in Water Resources 71, 177-185, doi:10.1007/s00193-013-0452-9.

Vousdoukas M. I., Voukouvalas E., Mentaschi L., Dottori F., Giardino A., Bouziotas D., Bianchi A., Salamon P., Feyen L., 2016. Developments in large-scale coastal flood hazard mapping, Natural Hazards and Earth System Sciences 16(8), 1841-1853, doi:10.5194/nhess-16-1841-2016.

Viero D.P., Defina A., 2016. Water age, exposure time, and local flushing time in semienclosed, tidal basins with negligible freshwater inflow, Journal of Marine Systems 156, 16-29, doi:10.1016/j.jmarsys.2015.11.006.

Viero D.P., Defina A., 2016. Renewal time scales in tidal basins: climbing the Tower of Babel. In "Sustainable Hydraulics in the Era of Global Change", Proceedings of the 4th IAHR Europe Congress, 27-29 July 2016, Eds: Erpicum et al., 338-345, Liege, Belgium.

Woodruff J. D., Irish J.L., and Camargo S.J., 2013. Coastal flooding by tropical cyclones and sea-level rise, Nature 504, 44-52, doi:10.1038/nature12855.

---

## Author Comment (AC1) · 19 May 2017

Thank you for your valuable revision of the manuscript. We will address your comments.

---

## Referee Comment (RC2) · Anonymous Referee #2 · 22 Aug 2017

Research investigates coastal flooding in Mexico from storm tides.

I found the text / descriptions are confusing and could do with improvement please; e.g., if high-pressure cold front event induced flooding I would expect, due to the inverse barometer effect, that storm surge is small and driven exclusively by wind stress? By improving the text will reduce reader fatigue. As the writing needs much improvement to improve readability, it may be easiest to simply re-submit. Moreover, I have a number of concerns with the method and results that I think ought to be addressed before re-submission:

(1) Tide-surge interaction means that "wetting and drying" is likely to be extreme impor-
tant in the model - yet this is not discussed/perhaps not included in the model? (2) Why are waves included in the method if not included in the model for the 30 year run? This could be done easily uncoupled - as coupled modelling likely to not be necessary? (3) The resolution and time-step of the CFSR forcing data needs to be discussed (hourly, 3hourly etc) - including a sensitivity test please. (4) More model validation please: "In general, a good agreement can be seen for the sea surface elevation during the storms the Pearson correlation ranges from 0.78 to 0.87 and the root mean square error (RMSE) ranges from 0.11 to 0.17" This model validation seems very poor. What is the RMSE as a percentage of the signal (NRMSE) ? Especially as a micro-tidal site and only a few validation locations (and limited time-length) appears to be used for validation

---

## Author Comment (AC2) · 25 Aug 2017

Thank you for your revision of the manuscript. We will address your comments soon and send a revised version of the manuscript.

---

## Editor Comment (EC1) · P. Tarolli (Editor) · 31 Aug 2017

Dear authors,

I would recommend to improve the interactive discussion with a more detailed reply to the reviewers' comments. I will make my decision, only after such additional replies.

Best regards

Paolo Tarolli

2017-64, 2017.

---

## Author Comment (AC3) · 12 Sep 2017

**"The role of tidal modulation in coastal flooding on a micro-tidal coast during Central American Cold Surge events"**

by Wilmer Rey et al.

**ANSWER TO COMMENTS OF REVIEWER #1**

**Major points**

1) The English of the paper needs to be thoroughly revised. Many typos, awkward and incorrect sentences can be found in the text and should be fixed. The help of a native speaker could be beneficial.

> **Authors response:** You are right. The new version of draft paper was reviewed by a professional native English speaker.

2) The paper actually deals with coastal flooding in a specific location. It does not assess, in general terms, the role tidal modulation in coastal flooding. Please consider to revise the title, as tidal modulation is only one aspect of the overall process analyzed in the paper. I'm thinking of something as "Assessment of coastal flooding in the Yucatan coast during Central American Cold Surge Events".

> **Authors response:** We propose "Assessment of coastal flooding and associated hydrodynamic processes on the Yucatan coast during Central American Cold Surge Events".

3) The presentation of the paper is very confused. Different type of data, validation of model results and of input data (i.e., water residuals and wind data), characterization of CACS events and analyses of the entire 30-tears time series, etc. are mixed together. The Authors should re-order, and possibly shorten, the paper, which needs to be less dispersive to the reader attention. Rather than being a vast report of all the analyses carried out by the Authors, a scientific paper should lead the reader to clear

> **Authors response:** You are right. We shortened significantly the draft paper. Figures 2 and 3 were joined, lower panel of Figure 5, figure 8 and figure 9 and their respective paragraphs of explanation were removed. Figures 10 and 11 were joint in two panels of a single figure. Based on this, we tried to focus only on the assessment of the inundation threat from CACS. In addition, the pertinent references you suggested were included in the manuscript, except #5 of your list (see list at the end of this document).

4) The Introduction is too long; many specific information should be moved from Introduction to Section 2.

> **Authors response:** We moved information from Introduction to section 2 and indeed improved clarity and readability.

5) Although an interest topic per se, I do not understand the role of hydrogeology in coastal flooding. How can an aquifer discharge affect the sea level? This question needs to be clearly assessed (if aquifer discharge actually plays some roles), or otherwise, being not even mentioned in the paper.

> **Authors response**: We acknowledge your comment. We assessed the contribution of the Yucatan aquifer to the sea level during CACS passes over the Yucatan Peninsula through a constant discharge along the coast. The model results indeed suggest that this contribution is not relevant (although the total discharge of the aquifer is estimated to be 4900822 $m^3.d^{-1}$ along 208 km of coast line (Weidie, 1985)) and then, as you suggest, was removed from the manuscript. Maybe in future studies, this can be analyzed in more detail.

6) The "References" section contains many typos and missing/wrong information (the formatting of conference proceedings and book chapter has problems with the conference/book title and with the number of pages). Please check carefully all the details of each bibliographic item.

> **Authors response:** Thank you. This was done. However, there are some conference proceeding that do not have book title, such as:
>
> Rey, W., Salles, P., Mendoza, E. T., Trejo-Rangel, M. A., Zhang, K., Rhome, J. and Fritz, C.: Hurricane Storm Surge Risk Assessment for the Yucatan State Coastal Area, in Proc. 32st Conference on Hurricanes and Tropical Meteorology, p. 1, American Meteorology Organization, San Juan, Puerto Rico., 2016.
> Torres-Freyermuth, A., Puleo, J. A., Ruiz de Alegría-Arzaburu, A., Figlus, J., Mendoza, T., Pintado-Patino, J. C., Pieterse, A., Chardon-Maldonado, P., DiCosmo, N. R., Wellman, N., Garcia-Nava, H., Palemón-Arcos, L., Roberts, T., López-González, J., Bravo, M., Ojeda, E., Medellín, G., Appendini, C. M., Figueroa, B., González-Leija, M., Enriquez, C., Pedrozo-Acuña, A. and Salles, P.: Nearshore Coastal Dynamics on a Sea-Breeze Dominated Micro-Tidal Beach (NCSAL), in Proc. American Geophysical Union, AGU Fall Meeting Abstracts, San Francisco, USA., 2014.
> They are found on the following websites:
> - http://adsabs.harvard.edu/abs/2014AGUFMOS11A1266T
> - https://ams.confex.com/ams/32Hurr/webprogram/Paper293952.html

We used the Copernicus style with default values for the references.

**Minor points**

-Check for the presence of double consecutive spaces into text.

> **Authors response:** Ok. Done.

-The use of acronyms and abbreviations should be limited, as it makes difficult to follow the text for those readers that are not already familiar with them. Finally, make sure that all the abbreviations are properly introduced when they first appear.

> **Authors response:** Ok. Done.

-Abstract: the reason why hindcast sea level time series was used (i.e., the lack of Interactive measurements) has to be stated. Rather than reporting specific numerical data, please specify the locations object of the study (Progreso and Chelem lagoon) and clearly outline the analyses carried out and the main results.

> **Authors response:** You are right. We made the corresponding changes on it.

-page 1, line 11: an "…occurrence probability" cannot be performed.

> **Authors response:** That sentence was change to: "To diagnose the mechanisms controlling the water levels, the two worst storms in terms of maximum residual tide (Event A), and maximum water level (Event B) were identified".

-p. 1, l. 16: "inlet" of what?

> **Authors response:** We refer to "lagoon inlet". The study area included a coastal lagoon, this is mention on the new version of abstract.
>
> -p. 1, l. 17: "despite micro-tidal conditions" what does this means? What is the difference between the tide (mentioned just before) and these "micro-tidal conditions"?
>
> **Authors response:** the term "microtidal" refers to tidal ranges less than 2 m (see, e.g., Pugh, 1987). If needed, it can be removed. In that same sentence, the difference between tide (noun) and micro-tidal (adjective) is obvious for us.

-p. 1, l. 34: "passing over the GoM"

> **Authors response**: That paragraph was deleted based on your major point # 3.

-p. 2, l. 4: "methodology" pertains to the "study of methods", use "method" instead.

(see also p.6, l. 14).

>**Authors response:** That paragraph was deleted based on your major point # 3.

-p. 2, l. 14: replace "induced by…… on the sea surface and" with "enhanced by"

>**Authors response:** Ok. Done.

-p. 2, l. 15: Shorten the sentence as "Consider that the effect of pressure field is relatively small during high-pressure atmospheric systems as CACS (Flather, 2001)."

>**Authors response:** That sentence was change as "Considering that the effect of atmospheric pressure is relatively small (Massey et al., 2007) especially during CACS events, the storm surge is mainly due to the shear stress of the wind, principally in shallow waters in the coastal zone (Flather, 2001)".

-p. 2, l. 19: which currents?

>**Authors response:** littoral currents.

-p. 2, l. 22: ":flood hazard is defined…"

>**Authors response:** Ok, Done.

-p. 2, l. 23: "…and period; it depends on…."

>**Authors response**: "it" was added to the sentence

-p. 2, l. 24: delete "However"

>**Authors response:** Ok. Done.

-p. 2, l. 25: The year is missing in the reference to Dorrestein

>**Authors response:** Sorry, it is Dorrestein, 1961

-p. 3, l. 13: delete "However"

>**Authors response:** Ok, Done.

-p. 3, l. 28: "back-barrier lagoon of Chelem, behind Progreso"

>**Authors response:** Ok. Done.

-p. 3, l. 32: Start a new paragraph with "In terms of hydrogeology…" (see also major point n. 5)

>**Authors response:** You are right, all of this paragraph was removed based on your major point # 5.

-p. 3, l. 34: delete "in"

> **Authors response:** that sentence was removed based on your major point # 5.

-p. 4, l. 17: units: remove periods from within units' expressions (e.g., $m^3s$ instead of $m^3$ .s). Please check throughout the text.

> **Authors response:** Ok, period was removed.

-p. 4, l. 17: put a reference to Fig. 2 after "Holbox".

> **Authors response:** Ok. Done

-p. 4, l. 22: please define "HD"

> **Authors response:** Ok. Done.

-p. 4, l. 23: "shallow water equations", not "shallow waters equations"

-p. 4, l. 23) Authors response: Ok. Done.

-p. 4, l. 25-26: awkward sentence

> **Authors response:** That sentence was change to: "By means of the integration of the horizontal momentum equations and the continuity (1) equation over $h = \eta + d$, the following two-dimensional shallow water equations are obtained."

-Eq. (1), (2), and (3): What kind of discharge is S, whose units are 1/s? How are the components of the "lateral stress" evaluated? Are these Reynold/dispersion stresses?

> **Authors response:** S is a discharge in $m^3$ $s^{-1}$, but first vertically integrated, and then per unit area.
>
> "The lateral stress terms include viscous friction, turbulent friction and differential advection. These are estimated using an eddy viscosity formulation based on the depth-averaged velocity gradients."

-p. 5, l. 4: delete "studying". This sentence seems incomplete.

> **Authors response:** That sentence was change to: "The wave model used to compute the wave conditions and associated radiation stresses was the MIKE 21 third generation spectral wave (SW) model. This model has been used for several spectral wind-wave modeling applications (Strauss et al., 2007; Appendini et al., 2013, 2015)."

-p. 5, l. 10: After "wave action equation" please put a reference (bibliographic or to an equation reported in the paper).

**Authors response:** The DHI (2014b) reference was added to that sentence.

-eq. (4) is correctly written?

**Authors response:** Yes, it was literally copied from the Spectral Wave manual.

-p. 5, l. 15: the sentence ": the directionally …formulation" is duplicated

**Authors response:** Ok. Done.

-p. 5, l. 18: "as described in"

**Authors response:** Ok. Done.

-p. 5, l. 21: what's the meaning of "(10-10 km)"?

**Authors response:** Sorry, it is 10-100 km.

-p. 5, l. 22: "and both swell and combined…are not important"

**Authors response:** Ok, it was corrected.

-p. 5, l. 27: I don't see S in the equation

**Authors response:** The energy source term, $S$, represents the superposition of source functions describing several physical phenomena. "$S = S_{in} + S_{nl} + S_{ds} + S_{bot} + S_{surf}$" was added to that sentence.

-p. 6, l. 9: "as reported"

**Authors response**: Ok. Done.

-p. 6, l. 14: Does the last sentence refer to the previously described treatment of the boundary condition? In this case, this sentence should be moved before the description of the boundary condition.

**Authors response:** Ok, this sentence was moved before the description of the boundary condition.

-p. 6, l. 19: "according to Arcement and Schneider (1989)", "of the Yucatan sand"

**Authors response:** Ok. Done

-p. 6, l. 22: what is the result of the further calibration of Cd?

**Authors response:** The "further calibration" consisted on: (a) varying (up and down) the Lin and Chavas (2012) values and selecting the best combination that resulted in the smallest WSL error. As in that paper, the Cd used varies linearly with the wind speed, as in other studies (e.g., Bryant, K. and Akbar, M.: An Exploration of Wind Stress Calculation Techniques in

Hurricane Storm Surge Modeling, J. Mar. Sci. Eng., 4(3), 58, doi:10.3390/jmse4030058, 2016). If needed, we can include details in the manuscript.

**The following minor points (p. 6, l. 37 to p. 7, l. 16) correspond to a section of the manuscript that was removed, addressing your suggestion of shortening the paper (see "Major Point #3".)**

-p. 6, l. 37-ff: This paragraph should be reorganized. The risk of coastal flooding is only associated with the total sea level, not directly with the sea residual. Clearly, the analysis of the sea residual is crucial, e.g. in order to improve sea level forecasts, since the sea residual is affected by major uncertainties than the astronomical tide (e.g., Met et al., 2014).

**Authors response:** That paragraph was removed (see above)

-p. 7, l. 5-10: D1 and D2 are datasets, i.e. sets of data, but they are described as actions/procedures ("consisted in identifying", "consisted in adding"). In D1 the astronomical tide is removed, in D2 is added again…Please make the description of the two datasets clearer.

**Authors response:** That paragraph was removed (see above).

-p. 6, l. 13 (in deep is p. 7, l. 13) : datasets have to be denoted with D1 and D2, not with (a) and (b).

**Authors response:** That sentence was removed (see above).

-p. 6, l. 16 (in deep is p. 7, l. 16): "selected and then analyzed"

**Authors response:** That sentence was removed (see above).

-p. 6, l. 26 (in deep is p. 7, l. 26): "At the peak". Remove the comma after "were"

**Authors response:** Ok, corrected.

-p. 6, l. 31 (in deep is p. 7, l. 31): "while" is a temporal expression, use "whereas for Event B.." instead. In addition, "closer to the normal to the coast".

**Authors response:** Ok, corrected.

-p. 8, l. 15-16: It is not clear to me how this goal was pursued. By shifting the astronomical tide for the entire 30-years' time series?

**Authors response:** No, only for Event A.

-p. 8, l. 27-ff: As for what I understand, a hypothetical scenario (TSSE) is compared with a measured (reanalyzed) wind field. Does this make sense?

**Authors response:** We were trying to compare/correlated the increase of the sea level with the increase of the offshore wind speed. However, that paragraph was removed from the draft based on the major point # 3.

-p. 9, l. 27 Figure 10, not Figure 9.

**Authors response:** That sentence and figure were removed from the draft addressing your suggestion of shortening the paper (see "Major Point # 3.")

-sect. 4 and sect. 5 are quite long. I suggest a sensibly shortening of these sections.

**Authors response:** These sections were shortened significantly as you suggested.

-p. 14, l. 11: "consist in using the… assuming that…and performing…"

**Authors response:** Ok. Done.

-p. 14, l. 16: "events"

**Authors response:** Ok. Done.

-p. 14, l. 23: A study cannot perform anything…

**Authors response:** Right, it was change to: "This study has developed a thirty-year sea-level hindcast."

-p. 14, l. 24: "to identify extreme water levels and characterize their probability of occurrence using…."

**Authors response:** Corrected.

-p. 14, l. 27: "different". "conditions", not "configurations".

**Authors response:** Corrected.

-p. 14, l. 31: The fact that an area is more populated cannot be a cause of more Flooding…..Rather, it can cause greater damages…

**Authors response:** Right, it change by "Since the wind stress over the lagoon was stronger for Event A, this event caused larger flooding than Event B over the back barrier lagoon of Progreso."

-p. 14, l. 33: "producing large set-up"; and waves? See, .e.g., Carniello et al. (2005).

**Authors response:** No, inside this small lagoon, the waves are not large (for the Event A, the biggest waves did not reached highs of 1 m. inside the lagoon).

-p. 14, l. 33: "Chelem lagoon". "were" in place of "occurred".

**Authors response:** Ok. Done

-p. 14, l. 34-35: "The passage of CACS, besides affecting water exchange with the sea and renewal dynamics inside the Chelem Lagoon (Viero & Defina, 2016a,b), is show to produce significant wind and wave set-up, characterized by nonlinear interactions between meteorological forcings and the astronomic tide.

**Authors response:** We totally agree about this. Your references were included.

-p. 15, l. 1: "Based on modeling results from…"

**Authors response:** "modelling" was added.

- p. 15, l. 2: "total flooded area"

**Authors response**: flooded was added.

-p. 15, l. 5: delete "is"

**Authors response:** "is" was added.

-p. 15, l. 6: awkward (and quite obvious) sentence.

**Authors response:** The sentence was change to: "However, the maximum flooding occurs when the CACS peak coincides with rising tide near zero level or high tide (TS4 and TS1 scenarios)". We do not consider that this is obvious, in particular during rising tide (near MWL).

-p. 15, l. 14: "storm surge, and set-up due to both wind and wave".

**Authors response:** Corrected

-Figure 3: I suggest putting the text in magenta on a white box to improve readability.

**Authors response:** Figure 2 and Figure 3 were joined.

-Figure 9, top panel: change the labels "Progreso" and "Progreso" with "Wind speed" and "Residual tide".

**Authors response:** this figure was removed based on your "major point #3".

-Figure 9, bottom panel: as for the top panel, labels should indicate the kind of data, not the location. "Reading (1992) method" is redundant here.

**Authors response:** this figure was removed based on your "major point #3".

-Additional References Interactive.

**Authors response:** Thanks for these suggestions.

1. Carniello L., Defina A., Fagherazzi S., D'Alpaos L., 2005. A combined wind wave-tidal model for the Venice lagoon, Italy. Journal of Geophysical Research – Earth Surface, 110, F04007, doi:10.1029/2004JF000232.

2. Mel R., Viero D.P., Carniello L., Defina A., D'Alpaos L., 2014. Simplified methods for real-time prediction of storm surge uncertainty: The city of Venice case study, Advances in Water Resources 71, 177-185, doi:10.1007/s00193-013-0452-9.

3. Vousdoukas M. I., Voukouvalas E., Mentaschi L., Dottori F., Giardino A., Bouziotas

4. D., Bianchi A., Salamon P., Feyen L., 2016. Developments in large-scale coastal flood hazard mapping, Natural Hazards and Earth System Sciences 16(8), 1841-1853, doi:10.5194/nhess-16-1841-2016.

5. Viero D.P., Defina A., 2016. Water age, exposure time, and local flushing time in semienclosed, tidal basins with negligible freshwater inflow, Journal of Marine Systems 156, 16-29, doi:10.1016/j.jmarsys.2015.11.006. Viero D.P., Defina A., 2016. Renewal time scales in tidal basins: climbing the Tower of Babel. In "Sustainable Hydraulics in the Era of Global Change", Proceedings of the 4th IAHR Europe Congress, 27-29 July 2016, Eds: Erpicum et al., 338-345, Liege, Belgium.

6. Woodruff J. D., Irish J.L., and Camargo S.J., 2013. Coastal flooding by tropical cyclones and sea-level rise, Nature 504, 44-52, doi:10.1038/nature12855.

---

## Author Comment (AC4) · 15 Sep 2017

**"The role of tidal modulation in coastal flooding on a micro-tidal coast during Central American Cold Surge events"**

by Wilmer Rey et al.

**ANSWER TO COMMENTS OF REVIEWER #2**

**Referee#2:** Research investigates coastal flooding in Mexico from storm tides. I found the text/descriptions are confusing and could do with improvement please; e.g., if high-pressure cold front event induced flooding I would expect, due to the inverse barometer effect, that storm surge is small and driven exclusively by wind stress? By improving the text will reduce reader fatigue. As the writing needs much improvement to improve readability, it may be easiest to simply resubmit. Moreover, I have a number of concerns with the method and results that I think ought to be addressed before resubmission:

Authors response: Thanks for your comment. On p. 2, I. 15 we mentioned something about it. However, we made some changes in that paragraph. The new paragraph says: "The storm surge is enhanced by the wind shear stress on the sea surface and perturbations in the atmospheric pressure (Lin and Chavas, 2012). Since the inverse barometer effect contribution to storm surge is small during low pressure storm systems (Massey et al., 2007), the storm surge is mainly driven by wind stress especially in shallow waters in the coastal zone (Flather, 2001). Considering that CACS are high-pressure systems, the storm surge is essentially driven by the direct wind effect". The effect of the pressure is relatively small; the rule of thumb is that for every 1 mb drop in pressure there is a 1 cm rise in ocean surface level (Massey et al., 2007). Besides, looking into the Figure 9 (removed from the paper based on major point # 3 of reviewer 1), it is shown that the peak of the residual tide occurs after the peak of the wind intensity (roughly 2 h) and after the minimum pressure (roughly 6 h). It means that the maximum residual occurred when the atmospheric pressure was around 1013 hPa (neutral pressure). By the time the CACS high pressure reached the Peninsula, the residual tide had already decreased. We assumed that this behavior might happen most of the time when CACS reach the Peninsula. In conclusion, the CACS storm surge is mainly driven by the direct wind effect.

On the other hand, we shortened the draft paper significantly and had it reviewed by a technical native English speaker reviewer (please see the answer to reviewer 1 for major point #3).

**Referee#2:** (1) Tide-surge interaction means that "wetting and drying" is likely to be extreme important in the model - yet this is not discussed/perhaps not included in the model?

**Authors response:** Wetting and drying are included in the model following the work by Zhao et al.(1994) and Sleigh et al. (1998). The user predefines the wetting and drying values so that the elements/cells are considered in the calculation only if the wetting threshold is surpassed. The elements/cells are removed from the calculations when the depth goes below a certain threshold, so that the momentum fluxes are set to zero and only considers the mass fluxes. The depth in each element/cell is monitored, and the elements are classified as dry, partially dry or wet. Besides, the element faces are monitored to identify flood boundaries.

The MIKE 21 wetting-and-drying algorithm performs skillfully inland to the east part of the back-barrier lagoon at Progreso. The inundation area calculated for Event A was of 8 and 149 blocks for the sea and lagoon side, respectively, as shown in Table 1 (blocks of Progreso affected as a function of tidal phase during Event A). We have included this on the discussion part.

**Referee#2:** 2) Why are waves included in the method if not included in the model for the 30 year run? This could be done easily uncoupled - as coupled modelling likely to not be necessary?

**Authors response:** The 30-year sea level hindcast was developed as the basis for the extreme level analysis, which is not possible from measurements due to the lack of tide gauge records. Unfortunately, the computational cost prohibits the modeling of coupled waves and hydrodynamics for that long period. For instance, for a given period of 3 weeks, and the computational domain is shown in Figure 2, the computational time for the uncouple model was 12 h, but increased drastically (up to 2 weeks) for the coupled model version (with waves). Given the computational resources available at that moment, it was not possible to carry out a sea level hindcast including waves. Nevertheless, we considered necessary to do coupled modeling to include wave setup, and wave-current interaction, in the particular cases presented in the study.

We then considered that to know the importance of taking into account the wave setup contribution on the total flood, the two worst storms in terms of maximum residual tide (Event A), and maximum water level (Event B) were chosen to assess the inundation threat on Progreso by means of running the hydrodynamic model in a coupled model. From this, we concluded that at least for Event A the wave set-up was significant given the no linear interaction wave-currents, which induced a relevant wave setup on the Chelem lagoon.

However, the wave setup contribution is usually not taken into account for some inundation modelers mainly because of the computational time cost and because most of the time it is not comparable with the storm surge contribution.

**Referee#2:** 3) The resolution and time-step of the CFSR forcing data needs to be discussed (hourly, 3hourly etc) - including a sensitivity test please.

**Authors response:** Thank you for the question. On p.6 I. 12 something related to this topic is mentioned, and we made some changes in that paragraph. The new version says "On the surface the model was forced with wind and pressure fields from the CFSR database, which has a global atmospheric resolution of 38 km (T382) with 64 levels extending from the surface to 0.26 hPa. The global ocean resolution is 0.25° at the equator, extending to 0.5° beyond the tropics, with 40 levels from the surface to a depth of 4737m. NCEP (National Centers for Environmental Prediction) has created time series products at hourly temporal resolution by combining either 1) the analysis and one- through five-hour forecasts, or 2) the one- through six-hour forecasts, for each initialization time. When using this data product, it has to be kept in mind that only the 0000, 0600, 1200, and 1800 UTC fields are actually analysis, while the in-between hourly data are model forecast. NCEP only created time series for parameter/level combinations that were thought to be most useful to users (Saha et al., 2010, 2014). Time series that do not exist in this dataset can be created from the full 6-hourly products dataset.

Given that the spatial resolution of the CFRS grid is not regular, and the hydrodynamic model only accepts wind and pressure data varying in space from a regular grid, CFSR wind and pressure fields were linearly interpolated from a T382 Gaussian grid resolution to a regular grid with spatial resolution of 0.3125°, which is coincident with the longitude of the T382 grid

and close in latitude for the Gulf of Mexico. We assumed this resolution to be adequate to reproduce the CACS storm surge based on the work of Appendini et al. (2013), who showed that the resolution of NCEP/NCAR, ERA-interim and NARR is sufficient for wave modeling of CACS over the Gulf of Mexico. Indeed, given that CFSR data is superior to the above NCEP reanalyses regarding (a) finer resolution, (b) advanced assimilation scheme as well as (c) atmosphere-land-ocean-sea ice coupling, it is expected to be a good compromise for this application. Moreover, the hourly resolution of CFSR allow this dataset to capture extremes, such as storm peak, which other reanalyses may miss, according to Sharp et al. (2015), who found a good correlation between the hourly CFSR dataset and both onshore and offshore in situ measurement for the U.K. For instance, NCEP FNL (Final), ECMWF ERA-Interim (European Centre for Medium Range Weather Forecasts e European Reanalysis) and NCEP-NCAR (National Centre for Atmospheric Research) provide data at 6 hourly intervals (Jørgensen et al., 2005), which may not be too long to capture storm peaks, and from that maximum flooding areas.

On the other hand, the MIKE 21 hydrodynamic model uses a dynamic time step to optimize simulation speed while ensuring stable model runs. Hence, the time step may change during the simulation (large time step under calm conditions, smaller time step when flow becomes stronger). The user is allowed to set the minimum and maximum time step in the model setup. The actual dynamic time steps used are found to be in the range from 5 to 7.5 s. Then, since the time step for the CFSR is 1 h (three orders of magnitude longer than the hydrodynamic model time step), the hydrodynamic model interpolates the CFSR data linearly to its own time step.

**Referee#2:** (4) More model validation please: "In general, a good agreement can be seen for the sea surface elevation during the storms the Pearson correlation ranges from 0.78 to 0.87 and the root mean square error (RMSE) ranges from 0.11 to 0.17" This model validation seems very poor. What is the RMSE as a percentage of the signal (NRMSE) ? Especially as a micro-tidal site and only a few validation locations (and limited time-length) appears to be used for validation.

**Authors response:** We acknowledge your comment. First, as mentioned before, one of the main problems on the study zone is the lack of long tide gauge records. That is the main reason why we made the sea level hindcast. In fact, we only have raw tide gauge records for 5 years and for one location (Progreso port) in the study area. For the model validation, we presented two different events (see figure 6). We calibrated the model based on the Drag Coefficient Cd to reproduce the maximum sea level during the CACS passing (please see for more details the answer to Reviewer 1 for the minor point mentioned on p.6, l.22).

We acknowledge that use of the term "ranges" in the sentence:

"In general, a good agreement can be seen in the sea surface elevation during the storms the Pearson correlation ranges from 0.78 to 0.87 and the root mean square error (RMSE) ranges from 0.11 to 0.17"

is confusing. Probably, based on this, you suggest that the validation is poor. Let us explain that Figure 6 as follows:

For the event shown on the top panel of Figure 6, the Pearson correlation is 0.78 and the RMSE is 0.1 m., which corresponds to the 20.9 % and 16.6 % of the measured and modeled sea level range, respectively. For the other event shown in the lower panel of Figure 6, the Pearson correlation is 0.87 and the RMSE is 0.17 m., which corresponds to the 16.6 % and 18.3 % of the measured and modeled sea level range, respectively. Based on the above, we considered that model validation is acceptable. For instant, the SLOSH model, which is the

only model used by the National Hurricane Center (NHC) to provide real-time hurricane storm surge (Massey et al., 2007) for over two decades, the accuracy of the predicted surge heights is +/-20% when the tropical cyclone is adequately described (Jelesnianski et al., 1992).

**References cited:**

Appendini, C. M., Torres-Freyermuth, A., Oropeza, F., Salles, P., López, J. and Mendoza, E. T.: Wave modeling performance in the Gulf of Mexico and Western Caribbean: Wind reanalyses assessment, Appl. Ocean Res., 39, 20–30, doi:10.1016/j.apor.2012.09.004, 2013.

Flather, R. A.: Storm Surges, in Encyclopedia of Ocean Sciences, edited by J. H. Steele, S. A. Thorpe, and K. K. Turekian, pp. 2882–2892, Academic, San Diego, California., 2001.

Jelesnianski, C., Chen, J. and Shaffer, W.: SLOSH: Sea, lake, and overland surges from hurricanes, NOAA Tech. Rep. NWS 48, United States Dep. Commer. NOAA/AOML.Library, Miami, Florida, 71, 1992.

Jørgensen, H., Nielsen, M., Barthelmie, R. J. and Mortensen, N. G.: Modelling offshore wind resources and wind conditions, Roskilde, Denmark: Risø National Laboratory., 2005.

Massey, W. G., Gangai, J. W., Drei-Horgan, E. and Slover, K. J.: History of Coastal Inundation Models, Mar. Technol. Soc. J., 41(1), 7–17, doi:10.4031/002533207787442303, 2007.

Saha, S., Moorthi, S., Pan, H. L., Wu, X., Wang, J., Nadiga, S., Tripp, P., Kistler, R., Woollen, J., Behringer, D., Liu, H., Stokes, D., Grumbine, R., Gayno, G., Wang, J., Hou, Y. T., Chuang, H. Y., Juang, H. M. H., Sela, J., Iredell, M., Treadon, R., Kleist, D., Van Delst, P., Keyser, D., Derber, J., Ek, M., Meng, J., Wei, H., Yang, R., Lord, S., Van Den Dool, H., Kumar, A., Wang, W., Long, C., Chelliah, M., Xue, Y., Huang, B., Schemm, J. K., Ebisuzaki, W., Lin, R., Xie, P., Chen, M., Zhou, S., Higgins, W., Zou, C. Z., Liu, Q., Chen, Y., Han, Y., Cucurull, L., Reynolds, R. W., Rutledge, G. and Goldberg, M.: The NCEP climate forecast system reanalysis, Bull. Am. Meteorol. Soc., 91(8), 1015–1057, doi:10.1175/2010BAMS3001.1, 2010.

Saha, S., Moorthi, S., Wu, X., Wang, J., Nadiga, S., Tripp, P., Behringer, D., Hou, Y. T., Chuang, H. Y., Iredell, M., Ek, M., Meng, J., Yang, R., Mendez, M. P., Van Den Dool, H., Zhang, Q., Wang, W., Chen, M. and Becker, E.: The NCEP climate forecast system version 2, J. Clim., 27(6), 2185–2208, doi:10.1175/JCLI-D-12-00823.1, 2014.

Sharp, E., Dodds, P., Barrett, M. and Spataru, C.: Evaluating the accuracy of CFSR reanalysis hourly wind speed forecasts for the UK, using in situ measurements and geographical information, Renew. Energy, 77, 527–538, doi:10.1016/j.renene.2014.12.025, 2015.

Sleigh, P. A., Gaskell, P. H., Berzins, M. and Wright, N. G.: An Unstructured Finite Volume Algorithm for Predicting Flow in Rivers and Estuaries, Comput. Fluids, 27(4), 479–508, doi:10.1016/S0045-7930(97)00071-6, 1998.

Zhao, D. H., Shen, H. W., Q, T. G., Lai, J. S. and Tan, W. Y.: Finite-Volume Two-Dimensional Unsteady-Flow Model for River Basins, J. Hydraul. Eng., 120(7), 863–883, doi:10.1061/(ASCE)0733-9429(1994)120:12(1497), 1994.

---

## Author Response (AR2)

**I. RESPONSE TO REVIEWER 1**

Review of: "Assessment of coastal flooding and associated hydrodynamic processes on the Yucatan coast during Central American Cold Surge events" by Wilmer Rey et al. NHESS-2017-64

The manuscript has been significantly improved by the Authors. I have few minor points concerning the presentation and some editorial suggestions, as detailed in the following.

Minor points

1) In sect. 3.2.1, the boundary conditions prescribed to the hydrodynamic models should be better described.

Response: We acknowledge the reviewer's comment. The description of the boundary conditions has been improved in the revised manuscript (p 5, L. 33-35 and p 8, L. 5-20).

i) Please clarify the meaning of a "mean profile of the Yucatan current" at p 5, lines 33-ff. Is this a time-varying boundary condition? What is the "current variability" mentioned at line 34?

Response: the current mean profile is constant in time, varying in space. By "current variability" we mean the variability of the Yucatan current. This current mean profile is just an approximation of the Yucatan current as Enriquez et al. (2010) did. Therefore, we revised that sentence in the revised manuscript to:

"The Yucatan channel boundary was forced with a mean profile of the Yucatan current, constant in time and varying in space, based on the results reported by Abascal et al. (2003). Part of the Yucatan current has been attributed to mesoscale eddies, which are observed in the eastern Caribbean basin, the Cayman Sea, and western Caribbean passages (Athié et al., 2011)."

ii) A sea level equal to zero at the Campeche boundary seems to me a quite "strong" boundary condition, as the western boundary is not so far from Progreso. Did the Authors check for the sensibility of water levels in Progreso to this boundary condition? (e.g., by extending the model domain westward.)

Response: We conducted a sensibility analysis, using a small computation domain and increasing the size up to get acceptable modeled water levels in Progreso port as shown in Figure 5. In p7, I 2-4 it is mentioned that the mesh size used for the HD model is the result of a sensibility analysis of the domain size (Blain et al., 1994; Morey et al., 2006; Kerr et al., 2013) at which the model adequately reproduced the sea level recorded by a tide gauge at Progreso. In fact, the model results close to the Campeche boundary, located 180 km from Progreso, are not very reliable.

It is true that a bigger computational domain might improve the surge modeled since it would allow us to simulate remote mechanism for storm-induce sea level rise. In this sense, probably extending the model domain westward, the results might improve, especially for CAC-events coming from northwest. However, it would significantly increase the computational time to simulate the 30-years.

2) Which are the three CACS events used to calibrate the model (p.6, l.36)? How does the model perform in simulating the events used for calibration? Are these events different from those use to validate the model?

Response: The three CACS-events (not shown) used to calibrate the model are independent to those events used to validate the model (Figure 5). However, the Pearson correlation and RMS values for the events used to calibrate the model are similar to the ones showed in Figure 5.

3) The tide scenarios simulated within the Case 4 should be better described and explained. The bullet points (p.8, l.21-24) is unacceptably confusing. Please describe the three additional scenarios referring only to phase shift and, only after that, explain why these phase shifts were chosen.

Response: We agree with the reviewer and apologize for the inconvenience. We re-write this section and stated on the new sentence that the boundary condition for case 4 had already been mentioned in section 3.2.2. Besides, a new Table 1 was created to describe the four tide scenarios considered.

Editorial suggestions-p.2 I.28: change "flux velocity" to "flow velocity".

Response: Done.

-p.2 I.: change "this problem" to "this lack of data".

Response: Done.

-p.4, eq. (1), (2), (3). S is given in  $s^{-1}$  (not in  $m^3s^{-1}$ ) for unit consistency. Please describe correctly this term at lines 16-17.

Response: Fixed. However, based on suggestion from reviewer #1 we shorted the model description (Section 3.1) by removing all the equations.

-p.4, l.26: "and only mass fluxes are considered."

Response: Equation removed.

-p.5: eq (4) is not correct as parentheses are missing. It should read N( $\sigma$ ,  $\theta$ ) = 1/  $\sigma$  E( $\sigma$ ,  $\theta$ ). Furthermore, E is not defined in the text (add at line 3 "where the energy density spectrum, E, is related to").

Response: Equation removed.

-p.5, l.11-12: Add a reference to support the sentence "The decoupled parametric formulation ... are not important" or, alternatively, delete this sentence.

Response: The sentence has been removed.

-p.5, eq (5): replace the period after the differential operator with a central dot.

Response: Equation removed.

-p.5, l.17: the variable S has already been used to denote the source term in the basic equations of the hydrodynamic model. Another symbol must be used here.

Response: Equation removed.

-p.5, l.32: "uncoupled"-p.6, l.1: "...along the boundary; values were extracted from..."

Response: Ok, thanks.

-p.6, l.11-12: "When using this these data product, it has to be kept in mind that only the 00:00, 06:00, 12:00, and 18:00 UTC fields..."

Response: Reviewer's suggestion has been incorporated.

-p.6, l.19: "... CFSR data are superior" (the term "data" is plural). "... then above NCEP reanalysis regarding due to..."

Response: Fixed.

-p.6, I.25-26: awkward sentence...

Response: The sentence has been revised accordingly. The sentence now reads: "and NCEP-NCAR (National Centre for Atmospheric Research) provide data at 6 hourly intervals, which may not be too long to capture storm peaks (Jørgensen et al., 2005). Therefore, when using these wind fields as forcing in hydrodynamic models the maximum flooding areas may be underestimated."

-p.6, I.28: "...while ensuring numerical stability"

Response: Done.

-p.7, l.19-20: "the largest residual tide that, being the less predictable tidal component, is relevant to flood hazard prediction"

Response: Done.

-p.7, I.25-26: please identify the two CACS events reporting the dates/time periods.

Response: The sentences was changed to: "(a) the event with the largest residual tide (Event A, whose peak occurred during receding tide), which hit the

Peninsula from March 12, 1993 at 16:00 to March 13 at 23:00, and (b) the event with the largest sea surface elevation (Event B, whose peak occurred during rising tide), which occurred from December 25, 2004 at 15:00 to December 26 at 09:00."

-p.8, l.6: Delete the first and the third part of the sentence. "the hydrodynamic model was forced by wind, tides, and mesoscale currents."

Response: Done.

-p.9, l.12: maybe "annual" in place of "yearly"

Response: Done.

-p.9, l.31: "as it is the case for Progreso"; remove the following comma.

Response: Done.

-p.11, I.32-35: This paragraph should be moved within Section 5.1.

Response: The paragraph has been moved.

-p.12, l.27: I do not understand the meaning of "at the battery"

Response: The Battery Park is a located in the New York city. The text has been revised to make it clear.

-p.14, Conclusion: please mention the location of the study (Progreso, Chelem Lagoon) befor speaking of the "inlet" (l. 26).

Response: The location is not mention before referring to the inlet.

-p.14, l.26-27: consider deleting "of arrival".

Response: Done.

-p15, l.1-5: use present tenses, as you are speaking of hypothetic scenarios, as well as real, scenarios. Alternatively, at line 1, say "...are tidally averaged: in the simulated scenarios, the maximum...".

Response: Following the reviewer's suggestion the present tense are used now.

-p.25, l.5: delete "(a) after "correspond to".

Response: Done.

-Figure 7: enlarge "A" and "B" within the figure.

Response: Figure 7 has been revised as suggested.

-Figure 8, 9, and 10: please indicate the plotted variable (e.g., "storm surge", "wave set-up", etc.) within each panel of the figure in order to improve the readability.

Response: Figures 8, 9, and 10 have been revised accordingly.

**II. RESPONSE TO REVIEWER 3**

Review of: "Assessment of coastal flooding and associated hydrodynamic processes on the Yucatan coast during Central American Cold Surge events" by Wilmer Rey et al. NHESS-2017-64

The manuscript provides an interesting study on surge, tide and wave interactions on the Yucatan Peninsula. I haven't read the first manuscript, but based on the comments of the previous reviewers, I think that the readability of the manuscript has improved significantly. In the study many thing are covered, which makes the manuscript at some locations a bit scattered, for instance is the GEV study really needed for the main aim of the study. Nevertheless, the study has a clear aim, which is addressed in the manuscript. Most of my comments (but not all) are text related issues; Before final publication, I would therefore recommend minor revisions.

Response: We thank the reviewer for his/her fruitful comments that helped us to improve the manuscript. We have incorporated all of them in the revised manuscript.

Major comments:

**1 Overall the text is well readable. Some sentence are still a bit of or strangely formulated, especially in the beginning of the manuscript. Once started with a series (on one hand) this should be finished. The same holds for contradictions as "larger" (larger than what) and the use of "but". I've numbers some of the cases below, but I'm not a language editor, so this should also be check.**

Response: We acknowledge the reviewer for his/her suggestions. A significant effort has been devoted to improve the manuscript readability.

**2 For many term abbreviations are used. The readability would be increased is the use of abbreviations is limited to the essential and commonly used terms, and/ or is a list with abbreviations is included.**

Response: In the revised manuscript the number of abbreviations has been decreased with respect to the earlier manuscript version.

**3 There are a lot of side studies in the manuscript, is Section 3.1, the GEV analysis and Section 4.4 essential for you analysis? Some of those side studies make the manuscript a bit broad, and it lead away from the main focus of the document. This is unfortunate, since the results are interesting.**

Response: We understand the reviewer's concern. Therefore, on the one hand we shortened the model description (Section 3.1) by removing all the equations. On the other hand, we decided keeping the GEV analysis since it was used to select Events A and B. Similarly, section 4.4 is also keep to highlight the individual contribution by the astronomic, storm surge and wave set-up at the inlet.

Minor comments

1. Title: I suggest to make clear in the title that the Yucatan coast is located in Mexico. Maybe change Yucatan coast to the Southeast coast of Mexico

Response: We thank the reviewer for such suggestion. The title has been modified accordingly.

2. p 1 In 20: Mention that the Yucatan Peninsula is located in Mexico (our on the south east coast of Mexico), add this also to line 4 on p 3.

Response: Done.

3. p 1 ln 20: see #1 "On one hand", these words suggest that an "on the other hand" I present as well. I miss this in the text

Response: We agree. We changed that sentence to:" On the one hand, extreme meteorological phenomena such as..... On the other hand, the region is characterized by a wide and shallow continental shelf.

4. p 1 ln 27: see #1"less frequent" it is not state opposed to what it is less frequent.

Response: The sentence was change to: "However, these type of events have a low occurrence along the northern Yucatan coast."

5. p 1 ln 29: (0.15 events/year) 2 events is too little for statistics

Response: Agree. This part was removed.

6. p 1 ln 30: see #1 "more frequent" compared to what?

Response: "more frequently" was removed from that sentence.

7. p 1 In 33: "Given the above" this is a bit a random not here, above there is a list of rates of occurrence, in my view it is not resulting in the aim of the study. Maybe adding a sentence that states that CACS events are occurring more frequent than hurricanes and that therefore the focus is on CACS-events. On p2 line 25 the aim is state clearly

Response: "Given the above" was replaced by "However, less efforts have been devoted to investigate the hazards associated to CASC events even though the annual frequency of CASC events is higher than hurricanes in the northwest of the Yucatan Peninsula. Thus, this study focuses on the effect of CACS events on coastal flooding."

8. p 2 ln 21: "smallness"  $\diamond$  the limited wave set-up

Response: Done.

9. p 2 ln 29: the sea level rising speed  $\diamond$  the speed of sea level rise

Response: Done.

10. p 2 ln 29-32: see #1 this sentence is a bit of. Check this sentence.

Response: We changed that sentence to: "Flood hazard assessments are normally performed based on historic or synthetic flood data (Lin et al., 2010; Zachry et al., 2015). However, since wind reanalysis datasets have become available, such as the North American Regional Reanalysis (Mesinger et al., 2006) and the Climate Forecast System Reanalysis-CFSR (Saha et al., 2010), they have been used to force hydrodynamic models to generate sea level reanalysis."

11. p 2 ln 40-p3 line 1: aim is clear

Response: Ok.

12. p 3: In 9-10: Although it is probably true. It is a bit weird that the two numbers don't add up to 100%

Response: The reviewer is right. To avoid confusion the sentence has been changed to: "The northern Yucatan coast is mostly sandy (85% of its length), from which 67% is formed by coastal lagoons and barrier islands (Cinvestav, 2007)."

13. p 3 ln 19: wind wave ◊ wind-wave

Response: Ok.

14. p 3 ln 20: and  $\diamond$  while or remore "on the other hand"

Response: Ok.

15. p 3 ln 21: Is GoM a common abbreviation? For readability Golf of Mexico could also be used (same with continental shelf)

Response: We limited the number of acronyms in the manuscript to the essential. Therefore, they have been reduced with respect to the earlier version of the manuscript.

16. p 3 ln 26: remove or replace "on the other hand"

Response: Done

17. p 4 section 3.1: If this is a summary of the DHI-models, this can be moved to the supplementary material, or after a short summary reference can be made to existing literature. Is there anything new that the authors added to the mathematical formulation?

Response: We agree with the reviewer. Since we have not added any mathematical formulation the section to address the reviewer's comments we have significantly shorten this section by removing mathematical formulations and unnecessary material. Interest readers are referred to the manual and papers.

18. p 5 ln 29-31: Shorten argumentation. To reduce computational time or for computational efficiency will do. Furthermore, I think that 2 week computational time is duable, if the number of simulations is limited, but even with more simulations, the simulations could be run parallel on multiple computer (or a super-computer).

Response: Right. That's is why we only ran the uncouple model to generated the 30- years sea level reanalysis. The idea was to run the couple model for the 30 years period but based on the computational cost, and our available computational resources at that time, it was not possible.

19. p 6 ln 14: accepts?

Response: "accepts" was replace for "takes"

20. p 6 ln 35: This is not visible in the figure.

Response: Right, the citation to Figure 2 was removed

21. p 7 ln 24: To what extend is this related to the fact that tropical cyclones are not well represented (ln 18-19)?

Response: First, the wind reanalysis database used (CFSR) underestimate the wind fields during tropical cyclones (TC), but not for CACS-events. In this regard, the sea surface elevation reproduced by the hydrodynamic model during the pass of TC is underestimated but nor for CACS-events. That's is why the sea surface elevation generated by TC was removed for the entire time series. In this sense, the maximum sea surface elevation in this study might be generated by CACS-events or any other phenomenon that can generate surges on the Yucatan coast. From this study, we state that all the 30 maximum belonging to the 30 events chosen are caused by CACS.

22. p 7 ln 23: presented?

Response: "presents" was replaced by "has".

23. p 7 ln 35: remove "on the other hand"

Response: Done.

24. p 7 ln 38: azimuth?

Response: Yes, it is correct.

25. p 8 ln 25: Mention at one location that TS scenarios are combined with Case 1 and Case 4.

Response: Right, that sentence was changed to : These four "tide scenarios" were also used with Case 1 and Case 4 to study the variation in residual tide and maximum flood as a function of the astronomic tidal phase for Event A, respectively.

26. p 8 ln 25: Maybe a Table could help to given an overview on the executed simulations.

Response: We thank the reviewer for the suggestion. A table has been included.

27. p 8 ln 36: In the beginning MIKE21 seems a bit underestimating the water levels, is your model spin-up time sufficient?

Response: Right, at the beginning the simulated water level is underestimated but at the storm peak the result get better. The model was calibrated to capture storm peaks since these maxima were used to assess the extreme analysis. The spin-up time (not shown in the figure) used was enough.

28. p 9 ln 18: also define H and x

Response: Done.

29. p 9 ln 22-27: rephrase, the content should be first, supported by the Figure.

Response: The sentence has been rephrased.

30. p 9 ln 29-30: This is a good, and well known, point.

Response: Yes, this is a good point to discuss.

31. p 9 ln 30: Remove "below"

Author response: Ok

32. p 10 ln 18: Please clarify the change in longshore current from northwest to southeast, how is this related to the drive of water from north, northwest?

Response: To address the reviewer's comments we made some changes on that sentence. During normal weather conditions there is a predominant littoral current along the northern coast of the Yucatan Peninsula from east to west. However, during CACS events, both (i) winds (from the northwest and north) produce a large and northerly shear stress on the sea surface, and (ii) pressure gradients due to atmospheric pressure perturbations, drive water towards the Peninsula. As a consequence, the predominant longshore current switches direction from west to east and leads to an increase in the sea level along the northern Yucatan coast and hence inside the coastal lagoons, due to the orientation of the coast (see Figure 1)

33. p 10 ln 34-35: "than for" check sentence

Response: "for" was removed.

34. p 10 ln 39: Could this also be related to 1) a limited tidal range in the lagoon, or 2) the limited waterdepth of the lagoon, which makes the storm surge level are strongly correlated with wind speeds?

Response: We thank the reviewer for pointing out this. We changed that sentence to:

"This is related to a limited tidal range and shallow water depth in the lagoon, resulting in a strong correlation between the storm surge levels and the wind stress, as well as a stronger wave setup, resulting in larger flooded areas of Progreso."

35. p 11 ln 2: change "," to "."

Response: Ok, thanks.

36. p 11 ln 1-6: Would it also be possible to refer to flooded m2 of km2 instead of number of flooded grid boxes. Or maybe something else is mentioned with the "blocks" in this case, please explain

Response: The "blocks" refer to "city blocks", not grid boxes. This was clarified in the text.

Besides, the following sentence was added: "Since assessing structural affectation is one the main objectives of this research, the block was chosen as the unit to show the flood prone area with inhabitants. In this regards, quantifying areas without inhabitants are beyond of the aim of this study." That's is why a total flood prone area in square meters is not provided.

37. p 11: Section 4.4 is a rather technical sections. State in the Figure 11 (caption) what is the x-axis is representing. This would really help understanding the explanation in Section 4.4. It now state distance, distance relative to what point?

Response: In order to clarify this concern, we made some changes on that sentence. "The 9-km-long transect passes through the inlet and starts 1,000 m inside the lagoon (i.e., the coastline is at x=1,000m), the end is offshore as shown in **Error! Reference source not found.** (panel c, black line)."

38. p 11 ln 17: What do you mean with breaking point? (point before waves start to break?)

Response: Yes, you are right.

39. p 11 ln 27: use of breaking point?

Response: Yes.

40. p 12 ln 2: remove "remembering that"

Response: Ok, thanks.

41. p 12 ln 30: Just as a not, it is not only the phase of the tide, but at some locations it is on top of that strongly related to the present of spring or neap tide.

Response: Thanks for the comment.

42. p 13 ln 23: What is "longer" morphology

Response: By inlet morphology we mean depth and length. We changed that paragraph to: Malhadas et al. (2009) suggested that wave set-up height inside the lagoon depends not only upon offshore significant wave height, but also on tidal inlet morphology (mainly depth and length). These authors demonstrated by means of numerical solutions of simple idealized models that the deeper and shorter is the morphology, the more the wave set-up is reduced.

43. Figure 1: If possible add counties to the map.

Response: Thanks for the comment.

44. Figure 2: Maybe show the bathymetry of the small nested area in a different subplot. The topo-bathymetry is invisible with the grid plotted on top of it. Remove abbreviation from the caption. The caption should be readable on its own.

Response: a transparency was set to the map to make the bathymetric visible.

45. Figure 4: What do the abbreviations mean?

Response: The abbreviations are defined the first time they appear in the text.

46. Figure 7: The marks of A and B in the figure should be better visible

Response: Done

47. Figure 8: the labels of the color bar are difficult to read.

Response: Done

**Assessment of coastal flooding and associated hydrodynamic processes on the Yucatan-Southeast coastof, Mexico, during Central American Cold Surge events**

Wilmer Rey1, Paulo Salles2,3, E. Tonatiuh Mendoza2,3, Alec Torres-Freyermuth2,3, Christian M. Appendini2,3.

[revised manuscript text omitted]

where η is the surface elevation, h the water depth, d the still water depth, the reference density of water, t the time, x, y are the Cartesian coordinates, g the gravitational acceleration, S the magnitude of discharge per unit volume in m3s-1 due to point sources, which were first integrated vertically and then per unit area, , are the velocities at which water is discharged into ambient water, p the density of water, , the depth averaged velocity in the x, y directions, the atmospheric pressure, , are the components of bottom stress, ,,, are the components of radiation stress tensor, , are the components of surface wind stress, ,,, are the components of lateral stress, and f is the Coriolis parameter. The lateral stress terms include viscous friction, turbulent friction and the stress terms include viscous friction.

30 differential advection. These are estimated using an eddy viscosity formulation based on the depth averaged velocity gradients. Any variable with an overbar indicates a depth average value.

Wetting and drying are included in the hydrodynamic HD model following the work of Zhao et al. (1994) and Sleigh et al. (1998). This model has been successfully used in recent scientific studies (Strauss et al., 2007; Appendini et al., 2014; Meza-Padilla et al., 2015).

35 The user predefines the wetting and drying values so that the elements/cells are considered in the calculation only if the wetting threshold is surpassed. The elements/cells are removed from the calculations when the depth goes below a drying threshold, so that the momentum fluxes are set to zero and only considers the mass fluxes\_are considered. The depth in each element/cell is

monitored, and the elements are classified as dry, partially dry or wet. Furthermore, the element faces are tracked to identify flood boundaries

The wave model used to compute the wave conditions and associated radiation stresses was the MIKE 21 third generation spectral wave (SW)-model. This model has been used for several spectral wind-wave modeling applications (Strauss et al., 2007; Appendini

- 5 et al., 2013, 2015). This wave model is based on unstructured meshes, and simulates the growth, decay and transformation of windgenerated waves and swell in offshore and coastal areas. The SW wave model includes the wave growth by action of wind, nonlinear wave wave interaction, dissipation due to white capping, dissipation due to bottom friction, and dissipation due to depthinduced wave breaking, as well as refraction and shoaling due to depth variations, wave current interaction and the effect of timevarying water depth and flooding and drying (DHI, 2014a, 2014b).
- 10 The SW wave module is based on the wave action equation (DHI, 2014b) where the wave field is represented by the wave action density spectrum *N*, formulated in terms of the relative angular frequency  $\sigma$ , and the direction of the wave propagation  $\theta$ , where the energy density spectrum,  $E(\sigma, \theta)$ , is related to the wave action density spectrum by

$$N = E, \tag{4}$$

This wave model includes two different formulations: directionally decoupled parametric formulation and fully spectral formulation: The first is based on a parameterization of the wave action conservation equation. The parametrization is performed in the frequency domain by introducing the zeroth and the first moment of the wave action spectrum as dependent variables as described in (Holthuijsen et al., 1989) and the second formulation is based on the wave action conservation equation as described in (Komen et al., 1994) and (Young, 1999), where the directional frequency wave action spectrum is the dependent variable. Since the fully spectral formulation is used for wave growth, decay and transformation of wind generated waves and swell in offshore and coastal areas, this formulation was chosen for this study. The decoupled parametric formulations are used more for small scale transformation applications (less than 10 100 km) and when the developed seas dominate and both swell and combined sea/swell are not important. The wave action conservation equation is written in Cartesian coordinates as

- +∇.·= (5)
- where the action density is defined by *N*, *t* is the time, = is the propagation velocity of a wave group in the four dimensional phase
  space σ and θ. ∇ is the four dimensional differential operator in the σ, θ space. The energy source term, *SZ*, represents the superposition of source functions describing several physical phenomena *SZ* = + + + + where represents the generation of energy by wind, is the wave energy transfer due to non linear wave wave interaction, is the dissipation of wave energy due to white capping, is the dissipation due to bottom friction and is the dissipation of wave energy due to depth induced breaking (DHI, 2014b). For more detailed information about source terms, governing equation, time integration and model parameters, readers are referred to Sørensen et al. (2004) and to the scientific manual documentation for the spectral wave SW model (DHI, 2014b).

**3.2 Model setup**

**3.2.1 Hydrodynamic model setup**

[revised manuscript text omitted]